

# An ensemble machine learning model based on multiple filtering and supervised attribute clustering algorithm for classifying cancer samples

Shilpi Bose[1], Chandra Das[1], Abhik Banerjee[1], Kuntal Ghosh[2], Matangini Chattopadhyay[3], Samiran Chattopadhyay[4] and Aishwarya Barik[1]

[1] Department of Computer Science and Engineering, Netaji Subhash Engineering College, Kolkata, West Bengal, India
[2] Machine Intelligence Unit & Center for Soft Computing Research, Indian Statistical Institute, Kolkata, West Bengal, India
[3] School of Education Technology, Jadavpur University, Kolkata, West Bengal, India
[4] Department of Information Technology, Jadavpur University, Kolkata, West Bengal, India

Corresponding author
Shilpi Bose, shilpi.bose@nsec.ac.in

## ABSTRACT

**Background:** Machine learning is one kind of machine intelligence technique that learns from data and detects inherent patterns from large, complex datasets. Due to this capability, machine learning techniques are widely used in medical applications, especially where large-scale genomic and proteomic data are used. Cancer classification based on bio-molecular profiling data is a very important topic for medical applications since it improves the diagnostic accuracy of cancer and enables a successful culmination of cancer treatments. Hence, machine learning techniques are widely used in cancer detection and prognosis.

**Methods:** In this article, a new ensemble machine learning classification model named Multiple Filtering and Supervised Attribute Clustering algorithm based Ensemble Classification model (MFSAC-EC) is proposed which can handle class imbalance problem and high dimensionality of microarray datasets. This model first generates a number of bootstrapped datasets from the original training data where the oversampling procedure is applied to handle the class imbalance problem. The proposed MFSAC method is then applied to each of these bootstrapped datasets to generate sub-datasets, each of which contains a subset of the most relevant/ informative attributes of the original dataset. The MFSAC method is a feature selection technique combining multiple filters with a new supervised attribute clustering algorithm. Then for every sub-dataset, a base classifier is constructed separately, and finally, the predictive accuracy of these base classifiers is combined using the majority voting technique forming the MFSAC-based ensemble classifier. Also, a number of most informative attributes are selected as important features based on their frequency of occurrence in these sub-datasets.

**Results:** To assess the performance of the proposed MFSAC-EC model, it is applied on different high-dimensional microarray gene expression datasets for cancer sample classification. The proposed model is compared with well-known existing models to establish its effectiveness with respect to other models. From the experimental results, it has been found that the generalization performance/testing accuracy of the proposed classifier is significantly better compared to other

well-known existing models. Apart from that, it has been also found that the proposed model can identify many important attributes/biomarker genes.

## INTRODUCTION

Cancer is one of the most fatal diseases around the globe (*Tabares-Soto et al., 2020*; *Hambali, Oladele & Adewole, 2020*). According to the World Health Organization report, Cancer is marked as the second most deadly disease and an estimated 9.7 million deaths around the world in 2018 have occurred due to this signature disease (*Hambali, Oladele & Adewole, 2020*). Generally, one in every six deaths all over the world, occurs due to cancer. So, within 2,030, the number of new cancer patients per year will increase approximately by 25 million (*Hambali, Oladele & Adewole, 2020*; *NIH, 2019*). Although several advanced techniques are already developed for the detection of cancer, the proper prognosis of cancer patients, till date, is very poor and the survival rate is also very low (*Tabares-Soto et al., 2020*; *Hambali, Oladele & Adewole, 2020*; *Kourou et al., 2015*). It has been already found that for very accurate cancer sample classification or prediction, adequate information is not available from the clinical, environmental, and behavioral characteristics of patients (*Kourou et al., 2015*; *Hambali, Oladele & Adewole, 2020*; *Tabares-Soto et al., 2020*). Recently, due to different types of bio-molecular data analysis, several genetic disorders with different biological characteristics have been revealed which are very helpful for early identification and prognosis of cancer and also to discern the responses for different types of treatment (*Colozza et al., 2005*; *Greller & Tobin, 1999*; *Li, Xie & Liu, 2018*; *Liu et al., 2011*; *Pilling, Henderson & Gardner, 2017*; *Su et al., 2001*; *Swan et al., 2013*).

With the rapid advancements in genomic, proteomic, and imaging high-throughput technologies (*Colozza et al., 2005*; *Greller & Tobin, 1999*; *Li, Xie & Liu, 2018*; *Liu et al., 2011*; *Pilling, Henderson & Gardner, 2017*; *Su et al., 2001*; *Swan et al., 2013*), now it is possible to accumulate huge amount (in the order of thousands) of different bio-molecular information of patients. Using this huge amount of information, researchers have been trying to develop more advanced techniques for early detection and proper prognosis of cancer, and also to improve cancer therapy for improvement of patients' survival rate. To analyze this huge amount of information, lab-based approaches are not adequate as these methods are costly and time-consuming. So, computational or in-silico methods like statistical methods, machine learning, deep learning, etc. have been being used extensively in this field.

It is well-known fact that in cancer-causing cells, gene expression is either overexpressed or under expressed (*Tabares-Soto et al., 2020*). So, measurement of gene expression in cancer cells can give adequate information to improve cancer diagnostic procedures.

Nowadays, different developing countries have been using this procedure for cancer sample detection. It is already known that using DNA microarray technology it is possible to measure the expression level of a numerous number of genes for a single experiment/sample simultaneously. The outcome of DNA microarray technology is a gene expression data matrix. This matrix carries information about the expression level of a huge number of genes for a limited number of samples (such as diseased patient samples and normal samples). The presence of the limited number of samples in this data matrix is due to the lack of availability of samples. So, based on information of gene expression data matrix, cancer sample classification is one of the essential tasks in the field of cancer research (*Chin et al., 2016*; *Dashtban & Balafar, 2017*; *Ding & Peng, 2005*; *Elyasigomari et al., 2017*; *Furey et al., 2000*; *Golub et al., 1999*; *Nada & Alshamlan, 2019*; *Tabares-Soto et al., 2020*).

Using computational or in-silico approaches, gene expression-based cancer sample classification task has been reviewed extensively in different papers (*Chin et al., 2016*; *Dashtban & Balafar, 2017*; *Ding & Peng, 2005*; *Elyasigomari et al., 2017*; *Furey et al., 2000*; *Golub et al., 1999*; *Nada & Alshamlan, 2019*; *Tabares-Soto et al., 2020*). However, the main difficulties in the sample classification task arise due to several factors. First, in these data sets, a substantially small number of samples is available (generally in the order of hundreds) compared to the availability of a huge number of genes (generally in the order of thousands) (*Chin et al., 2016*; *Hambali, Oladele & Adewole, 2020*; *Nada & Alshamlan, 2019*). For sample classification, genes are treated as features/attributes. So, the high-dimensional gene space is an overhead for most classification algorithms. Second, only a very few genes are informative (differentially expressed) and the rest of the section is non-informative (noisy) (*Chin et al., 2016*; *Hambali, Oladele & Adewole, 2020*; *Nada & Alshamlan, 2019*) for sample classification and responsible for degrading the classifier's performance. Gene dimension reduction by identification of informative genes as biomarkers can improve the classification accuracy of classifiers. Apart from the improvement of classification accuracy, the identification of informative biomarkers (here, informative genes) has great prospects from a biomedical point of view. These are beneficial for finding the biological reason for a disorder, assessing disease risk, and developing therapeutic targets. The third problem arises due to the small sample size which creates an overfitting problem in classifier construction. Another problem that degrades classifier performance is the sample class imbalance problem. This problem occurs due to the presence of more instances/samples of one class (majority class) with respect to other class(es) (minority class) in a dataset.

A fairly large number of works have been already developed for sample classification. These works are divided into two categories. In the first category (*Chin et al., 2016*; *Hambali, Oladele & Adewole, 2020*; *Nada & Alshamlan, 2019*), the major emphasis is given to the selection of relevant genes for the reduction of feature space. Then based on this reduced feature space, predictive/classification accuracy of the samples is measured using different existing single classification models like naïve Bayes, support vector machine, relevance vector machine, K-nearest neighbor, decision tree, logistic regression, etc. As gene selection is a feature selection task, so based on feature selection techniques, these

methods are divided into different categories. These are (1) filter methods (2) wrapper methods (3) embedded methods and (4) hybrid methods. Before we mention the second category of classification methods, let us first elaborate on the first category methods one by one.

Filter methods (*Chin et al., 2016*; *Hambali, Oladele & Adewole, 2020*; *Nada & Alshamlan, 2019*) select a subset of features without taking any information from any classification model. These methods select features that are differentially expressed with respect to sample class labels. The filter methods rank individual features according to their class discrimination power based on some statistical score function and then select a number of high-ranked features to form a reduced and relevant feature subset. The popular statistical score functions used in filter methods are Fisher's score, Signal to Noise ratio (SNR), correlation coefficient, mutual information, Relief (*Das et al., 2019*), etc. Filter methods are computationally simple, fast, and unbiased in favor of any specific classifier as these methods do not consider any knowledge from any classifier at the feature selection phase. The drawback of filter methods is that the number of selected features is based solely on the trial-error method.

Wrapper methods (*Chin et al., 2016*; *Hambali, Oladele & Adewole, 2020*; *Nada & Alshamlan, 2019*), on the other hand, judge discrimination capability of a feature subset using classification error rate or prediction accuracy of a classifier as the feature evaluation function. It selects the most discriminative feature subset *via* minimizing the classification error rate or maximizing the classification accuracy of a classifier. The wrapper methods generally achieve better classification accuracy than the filter methods because the selection of feature subset is classifier-dependent. One drawback of these methods is that these are biased to used classifiers and another drawback is that these are computationally more expensive than the filter methods as generation of the best feature subset for the high-dimensional dataset is an NP-complete problem. Due to these reasons, these methods are not applicable for high-dimensional datasets.

In Embedded methods (*Chin et al., 2016*; *Hambali, Oladele & Adewole, 2020*; *Nada & Alshamlan, 2019*), the optimal feature subset is selected through the unique learning procedure of a specific classifier at the time of classifier construction. Actually, in these methods, the optimal feature subset selection part is embedded as part of classifier construction. These methods are faster than wrapper methods but are biased to the specific classifier. In embedded approaches, the feature selection process is specific for a particular classifier and is not applicable to other classifiers. These are also computationally expensive. Due to these reasons for high-dimensional datasets, these methods are not applicable. On the other hand, recently hybrid feature selection methods (*Chin et al., 2016*; *Hambali, Oladele & Adewole, 2020*; *Nada & Alshamlan, 2019*) are also developed. In hybrid methods, different category-based methods are combined to take advantage of all of these methods for improving classification accuracy.

Apart from these methods, clustering techniques (*Chin et al., 2016*; *Hambali, Oladele & Adewole, 2020*) are also used for feature selection purposes. Clustering techniques divide the data space in such a manner that objects in the same cluster are similar while in different clusters they are dissimilar. For the feature selection task, clustering methods

(famous as attribute clustering in feature selection domain) (*Au et al., 2005*) divide the features into several distinct clusters and then reduce the feature dimension by selecting a small number of significant features from each cluster. A lot of unsupervised gene (attribute) clustering algorithms (*Au et al., 2005*; *Chin et al., 2016*; *Hambali, Oladele & Adewole, 2020*) are already developed for this task. However, these methods are unsuccessful to find informative functional groups of genes for sample classification as in clustering genes, no supervised information from sample classes is considered (*Au et al., 2005*; *Chin et al., 2016*; *Hambali, Oladele & Adewole, 2020*). So, scientists have developed a number of supervised gene (attribute) clustering algorithms (*Dettling & Buhlmann, 2002*; *Hastie et al., 2000*; *Hastie et al., 2001*; *Maji & Das, 2012*) in which genes are grouped using supervised information from sample classes and a reduced gene set is formed *via* selecting the most informative genes from each cluster.

All the above-mentioned variants deliver comparable feature selection and classification accuracy. Quite often this type of classification models with only a few genes and with a limited number of training samples can classify the majority of training samples correctly, but the generalization capability of such classification models cannot be guaranteed (*Bolón-Canedo, Sánchez-Maroño & Alonso-Betanzos, 2012*; *Ghorai et al., 2011*; *Nagi & Bhattacharyya, 2013*; *Wang, 2006*, *Wang, Li & Fanget, 2012*; *Yang et al., 2010*). So, the most important task for a medical diagnosis system is to improve the classification accuracy of unknown samples (generalization performance) which cannot be solved by this type of classification model.

Apart from this problem, the microarray data is related to several uncertainties due to fabrication, hybridization, and image processing procedure in microarray technology. These uncertainties introduce various types of noise in microarray data. Due to the presence of these uncertainties with a limited number of training samples, the conventional machine learning approaches face challenges to develop reliable classification models.

To overcome the above-mentioned problems, it is therefore essential to develop general approaches and robust methods. In this regard, researchers are motivated to develop the second category-based model. These are the different robust ensemble classification models (*Bolón-Canedo, Sánchez-Maroño & Alonso-Betanzos, 2012*; *Ghorai et al., 2011*; *Nagi & Bhattacharyya, 2013*; *Osareh & Bita, 2013*; *Wang, 2006*; *Wang, Li & Fanget, 2012*; *Yang et al., 2010*) which can overcome small sample size problems and are capable of removing uncertainties of gene expression data.

Ensemble methods (*Dietterich, 2000*) are a class of machine learning technique which combines multiple base learning algorithms to produce one optimal predictive model. Ensemble classification model refers to a group of individual/base classifiers that are trained individually on the trained dataset in a supervised classification system and finally, an aggregation method is used to combine the decisions produced by the base classifiers. These ensemble classification models have the potential to alleviate the small sample size problem by applying multiple classification models on the same training data or on bootstrapped samples (sampling with replacement) of the training data to decrease the

chance of overfitting in the training data. In this way, the training dataset is utilized more efficiently, and as a consequence, the generalization ability is improved.

Although different category-based ensemble classification models exist in the literature but these ensemble models are not capable of addressing all the above-mentioned problems (small sample size, high dimensional feature space, and sample class imbalance problem) related to microarray data.

In this regard, here a new Multiple Filtering and Supervised Attribute Clustering algorithm-based ensemble classification model named MFSAC-EC is proposed. In this model, first, a number of bootstrapped versions of the original training dataset are created. At the time of the creation of bootstrapped versions, an oversampling technique (*Błaszczyński, StefanowskiŁ & Idkowiak, 2013*) is adopted to solve the class imbalance problem. For every bootstrapped dataset a number of sub-datasets (each with a subset of genes) are generated using the proposed MFSAC method. The MFSAC is a hybrid method combining multiple filters with a new supervised attribute clustering method. Then for every sub-dataset, a base classifier is constructed. Finally, based on the prediction accuracy of all these base classifiers of all sub-datasets for all bootstrapped datasets an ensemble classifier (EC) is formed using the majority voting technique.

The novelty of the proposed MFSAC-EC model is that here the emphasis is given simultaneously on the high dimensionality problem of gene expression data, small sample size problem as well as the class imbalance problem. All of these problems at the same time are not considered in any existing ensemble classification model. First of all, due to the use of bootstrapping method with a class balancing strategy, the proposed model can handle a small sample size and overfitting problem. Second, in MFSAC, different filter methods are used with their unique characteristics. So, different characteristics-based relevant gene subsets are selected *via* different filters to form different sub-datasets from every bootstrapped dataset. Finally, every gene subset is modified using a supervised attribute clustering algorithm. In this way, the high-dimensionality problem of gene expression data is handled here. Apart from this, from the MFSAC generated sub-datasets, the frequency of occurrence is counted for every gene and informative genes are ranked accordingly. The prediction capability of the proposed model is experimented with over different microarray datasets and compared with the existing well-known models. Experimental outputs demonstrate the superiority of the proposed model over existing models.

## MATERIALS & METHODS

The proposed MFSAC-EC model is composed of different filter score functions, a new supervised attribute clustering method, and an ensemble classification method. In the following subsections, first, a brief overview is given on different filter score functions and then the proposed MFSAC-EC model is described.

### Preliminaries

In this paper, a data set (here, a microarray gene expression data set) is represented by a data matrix, $K_{U \times V}$, with $U$ data objects (samples) and $V$ features (genes). The set of objects

or samples is represented as $E = \{E_1, E_2, \ldots, E_s, \ldots E_U\}$ while the set of genes is represented as $G = \{G_1, G_2, \ldots, G_t, \ldots G_V\}$. Here, each sample is a $V$-dimensional feature vector containing $V$ number of gene expression values. Similar way, every gene is a $U$-dimensional vector containing $U$ number of sample values. Here, $C_{U \times 1}$ is a class vector representing the associated class label for every sample. The class label is taken from a set $DC = \{d_1, d_2, \ldots, d_j, \ldots d_N\}$ with $N$ distinct class labels.

## Brief overview of filter score functions used in MFSAC

The filter score functions used in the proposed MFSAC-EC model are modified Fisher score (*Gu, Li & Han, 2011*), modified T-test (*Zhou & Wang, 2007*), Chi-square (*Das et al., 2019*), Mutual information (*Das et al., 2019*), Pearson correlation coefficient (*Leung & Hung, 2010*), SNR (*Leung & Hung, 2010*) and Relief-F (*Das et al., 2019*). A summary of these seven filters used in the MFSAC-EC model is given in the Table S1.

## Proposed MFSAC-EC model

In the proposed MFSAC-EC model, initially, bootstrapping (sampling with replacements) with a class balancing procedure of samples is applied on training dataset $K$ to create $D$ number of different bootstrapped versions from the training dataset. Here, every bootstrapped dataset with $U$ samples is formed by random sampling with replacements $U$ times from the original dataset $K$. After that oversampling procedure is applied to each minority class to achieve data balance. Oversampling consists of increasing the minority class instances by their random replication to exactly balance the cardinality of the minority and majority classes in each bootstrapped dataset. Due to oversampling each bootstrapped dataset will contain more instances than the original dataset.

The MFSAC method of the MFSAC-EC model, which is an integration of multiple filters and a new supervised attribute (gene) clustering method, is applied on every newly created bootstrapped ($BK_l$) training dataset. The proposed MFSAC method first calculates the class relevance score of every gene present in the bootstrapped training dataset using each filter score function ($FT_x$), $x = 1\ to\ 7$ mentioned above. Then for each filter score function, a sub-dataset $SD_{lx}$ with a gene subset ($GS_{lx}$) is created by selecting a predefined number (let $P$) of the most relevant genes from the full gene set $G$. So, $|GS_{lx}| = P$. After that on every gene subset ($GS_{lx}$) of every sub-dataset $SD_{lx}$, the SAC (Supervised Attribute Clustering) method is applied and a set of clusters $CGS_{lx}$ and corresponding cluster representatives (considered as modified features) are formed. Finally, $Q$ numbers of most relevant cluster representatives are selected as modified features and a reduced sub-dataset $RSD_{lx}$ of the sub-dataset $SD_{lx}$ is formed. How the SAC method works on $GS_{lx}$ of every sub-dataset $SD_{lx}$ is discussed below.

For any sub-dataset $SD_{lx}$, the SAC method starts by selecting the gene from the subset ($GS_{lx}$) with the highest $FT_x$ value. Let gene $G_{li} \in GS_{lx}$ with the highest $FT_x$ value be selected as the first member (let $FT_x(G_{li}, C) = A$) and it also becomes the initial cluster representative $R(R = G_{li})$) of the first cluster $C_1GS_{lx}$ and $G_{li}$ is deleted from $GS_{lx}$. In effect, $G_{li} \in C_1GS_{lx}$, and $GS_{lx} = GS_{lx} - \{G_{li}\}$ and so $FT_x(R, C) = A$. This cluster is then grown up in parallel with the cluster representative refinement process which is described

next. In this process, the gene (let $G_{lm}$) with next highest $FT_x$ value is taken from $GS_{lx}$ subset and is merged with the current cluster representative $R$. The merging is done in two ways. First, the expression profile of $G_{lm}$ is directly added with $R$ and a temporary augmented representative $TR^+$ is formed and its $FT_x$ value (let $B_1$) is calculated. The second one is that the sign-flipped value of the expression profile of $G_{lm}$ is added with $R$ and another temporary augmented representative $TR^-$ is formed and its $FT_x$ value (let $B_2$) is calculated. If $FT_x(TR^+, C) \geq FT_x(TR^-, C)$ that is $B_1 \geq B_2$ then $TR^+$ is chosen else $TR^-$ is chosen. Let $TR^+$ is chosen. Now if $FT_x(TR^+, C) > FT_x(R, C)$ then $R = TR^+$ otherwise, $R$ is unaltered. Similar way if $TR^-$ is chosen and if $FT_x(TR^-, C) > FT_x(R, C)$ then $R = TR^-$ otherwise, $R$ remains unchanged. If $R$ is modified then the gene $G_{lm}$ is included in the cluster and $G_{lm}$ is deleted from $GS_{lx}$. In effect, $G_{lm} \in C_1GS_{lx}$, and $GS_{lx} = GS_{lx} - \{G_{lm}\}$. So, the next chosen gene is included in the current cluster if it improves the class relevance value of the current cluster representative. The merging process is described in Fig. 1.

Here g0 represents the current cluster representative $(R)$ and its class relevance score $((FT_x, R)$, here Pearson score), is shown. Now among all the genes g1, g2, g3, g4, and g5, the Pearson score of g1 is the highest. So, g1 is chosen for the merging process. Then g1 is added with $R$ to create the temporary augmented representative $(TR^+ = R + g1)$ and also its sign-flipped value is added with the $R$ to form the temporary augmented representative $(TR^- = R - g1)$. The Pearson score of $TR^+$ is greater than the Pearson score of $TR^-$, so $TR^+$ is chosen. Now the Pearson score of $TR^+$ is greater than the Pearson score of $R$, so $TR^+$ is considered as the current cluster representative and $R = TR^+$. This process is continued for all other genes. Now, g3 is chosen as it is the gene with the next highest Pearson value. g3 and its sign-flipped value are added individually with current cluster representative $R$ to form $TR^+ = R + g3$ and $TR^- = R - g3$ respectively. In this case, Pearson score of $TR^-$ is greater than the Pearson score of $TR^+$. So, $TR^-$ is chosen. Then Pearson score of $TR^-$ is Checked with the Pearson score of $R$ and here Pearson score of $TR^-$ is greater than the Pearson score of $R$. So, $TR^-$ is considered as current cluster representative and $R = TR^-$. In this way, cluster representative is refined. This process is repeated for every member of $GS_{lx}$ subset.

After the formation of the first cluster and its corresponding augmented representative, $R$ is assigned to $AR_{lx1}$ that means $AR_{lx1} = R$, and the supervised clustering process is repeated to form the second cluster with the gene (let $G_{lz}$) with next highest $FT_x$ value from $GS_{lx}$ subset. In this way a set of clusters $CGS_{lx} = \{C_1GS_{lx}, C_2GS_{lx}, \ldots, C_kGS_{lx}, \ldots\}$ and their corresponding augmented cluster representatives $AR_{lx} = \{AR_{lx1}, \ldots, AR_{lxk}, \ldots\}$ are formed. After that $Q$ number of most powerful augmented cluster representatives are chosen (as modified features) according to their $FT_x$ value from the generated clusters and with these $Q$ number of modified features, a reduced sub-dataset $RSD_{lx}$ of sub-dataset $SD_{lx}$ is formed.

In this way, for every bootstrapped version $(BK_l)$ of the training dataset, seven number of $RSD_{lx}$ sub-datasets are created and for every $RSD_{lx}$ an individual classifier is constructed using any existing classifier and finally, an ensemble classifier (EC) is formed

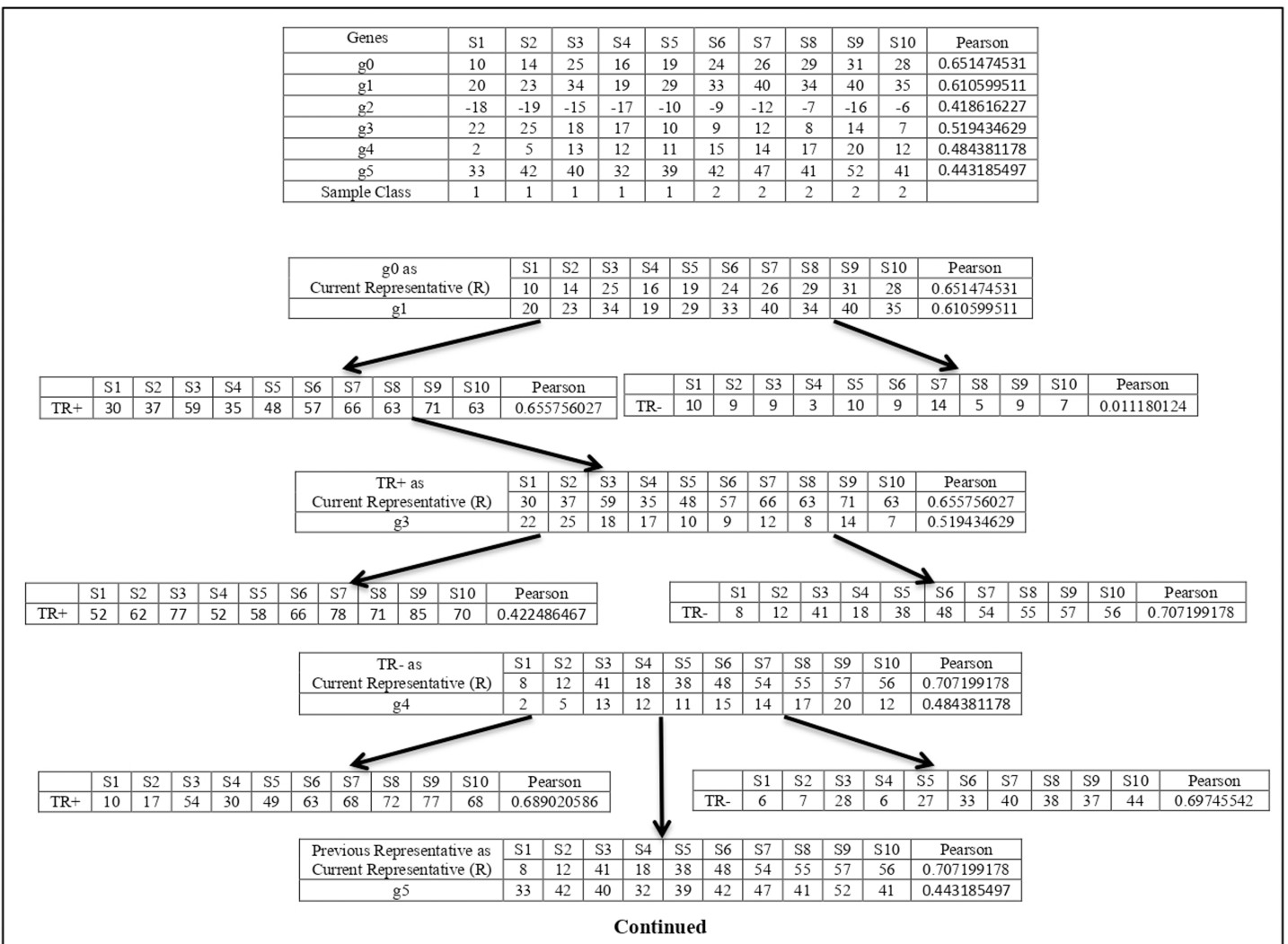

**Figure 1 Cluster representative refinement procedure.** Each row of the table represents the gene with its class relevance value in terms of Pearson correlation coefficient with respect to sample class row. TR⁺ and TR⁻ represent the augmented gene with their class relevance score in terms of Pearson correlation coefficient with respect to sample class row.                

by combining all these classifiers of all bootstrapped versions using the majority voting technique. To classify every sample using this ensemble classifier, each classifier votes or classifies the sample for a particular class, and the class for which the highest number of votes is obtained is considered as the output class.

## MFSAC method based informative attribute ranking

For every gene (feature/attribute), the frequency of occurrence that means the total number of times it appears in all sub-datasets generated by the MFSAC method for all bootstrapped versions is calculated. Then according to their frequency of occurrence, those genes are ordered or ranked. The top-ranked genes with the highest occurrence frequency are considered the most informative cancer-related genes.

---

### Algorithm: MFSAC-EC

**Input:** A $K_{U \times V}$ data matrix (here, gene expression data matrix) containing $U$ number of data objects (here, cancer samples) and $V$ number of attributes (here, genes).

**Output:** An ensemble classifier MFSAC-EC is formed to classify test samples. From MFSAC generated sub-datasets, informative genes are selected according to their rank. Every gene is ranked according to its frequency of occurrence.

**Definitions:**

$E = \{E_1, E_2, \ldots, E_s, \ldots E_U\}$ is the set of objects or samples of $K_{U \times V}$ data matrix. Every sample $E_s$ is a $V$ dimensional vector.

$G = \{G_1, G_2, \ldots, G_t, \ldots G_V\}$ is the set of features or genes of $K_{U \times V}$ data matrix. Every gene $G_t$ is a $U$ dimensional vector.

$BK = \{BK_1, BK_2, \ldots, BK_l, \ldots, BK_D\}$ is a set of the bootstrapped version of the original training dataset. In every bootstrapped dataset the number of samples varies from the original dataset but the number of features is the same as the original dataset.

$C_{U \times 1}$ is a class vector representing the associated class label for every sample. For a data matrix $N$ distinct class labels exist and class labels are taken from a set $DC = \{d_1, d_2, \ldots, d_k, \ldots d_N\}$.

$FT_x(G_t, C)$ is $x^{th}$ filter score function which returns the class relevance value of $G_t$ gene with respect to class vector $C$ using $FT_x$ score function, for $x = 1$ to $7$ as $7$ represents the total number of filtering score functions used here.

$GS_{lx}$ ($GS_{lx} = P$) is a set of top-ranked genes of $G$ selected using $FT_x$ score function and $SD_{lx}$ is corresponding sub-dataset of $BK_l$. Here $SD_{lx}$ is a data matrix containing $P$ number of genes.

$CGS_{lx} = \{C_1 GS_{lx}, C_2 GS_{lx}, \ldots, C_k GS_{lx}, \ldots\}$ and $AR_{lx} = \{AR_{lx1}, \ldots, AR_{lxk}, \ldots\}$ are the set of clusters and corresponding cluster representatives respectively generated from the corresponding subset $GS_{lx}$ of $SD_{lx}$. Here every $AR_{lxk}$ is a vector.

$TR^+$, $TR^-$, R are vectors similar to a gene vector.

$RSD_l = \{RSD_{l1}, RSD_{l2}, \ldots, RSD_{lx}, \ldots RSD_{l7}\}$ is a set of sub-datasets each containing $Q$ number of most relevant cluster representatives formed for every bootstrapped dataset $BK_l$.

$CF_l = \{IC_{l1}, IC_{l2}, \ldots, IC_{lx}, \ldots IC_{l7}\}$ is a set of classifiers formed for every bootstrapped dataset.

1. Create $D$ number bootstrapped version of training dataset $K$.

2. For Every bootstrapped dataset $BK_l$ repeat step 3

3. Repeat for $x = 1$ to $7$

    A. Repeat for $t = 1$ to $V$

        a) Calculate class relevance score $FT_x(G_t, C)$ of $G_t$ gene, where $G_t \in G$, with respect to class vector $C$

    B. Select $P$ number of top-ranked genes from $G$ based on $FT_x$ score function and form $GS_{lx}$ gene subset with corresponding $SD_{lx}$ sub-dataset

    C. Set $k = 0$

    D. Repeat until $GS_{lx} = \varnothing$

        a) Set $k = k + 1$

        b) Set $AR_{lxk} = 0$, $R = 0$, and $i = 0$

        c) Select the gene (let $G_{li}$) whose $FT_x$ score value is maximum among all genes of $GS_{lx}$ and set $R = G_{li}$

        d) Add $G_{li}$ to $C_k GS_{lx}$, and delete $G_{li}$ from $GS_{lx}$

        e) Set count =1

        f) Repeat for $j = 1$ to $|GS_{lx}|$

            I. Compute first augmented representatives $TR^+$ by adding $G_{lj} \in GS_{lx}$ with R that means $TR^+ = R + G_{lj}$

            II. Compute second augmented representatives $TR^-$ by adding sign-flipped version of $G_{lj} \in GS_{lx}$ with R that means $TR^- = R - G_{lj}$

            III. Compute class relevance value $FT_x(TR^+, C)$ and $FT_x(TR^-, C)$ using $FT_x$ score function

            IV. If $FT_x(TR^+, C) \geq FT_x(TR^-, C)$ then

                If $FT_x(TR^+, C) > FT_x(R, C)$ then

| Algorithm: (continued) |
| --- |

        ● Set$R = R + G_{lj}$ and add $G_{lj}$ to $C_k GS_{lx}$ and delete $G_{lj}$ from $GS_{lx}$

        ● count = count +1

   V. If $FT_x(TR^-, C) > FT_x(TR^+, C)$ then

If $FT_x(TR^-, C) > FT_x(R, C)$ then

        ● Set$R = R - G_{lj}$ and add $G_{lj}$ to $C_k GS_{lx}$ and delete $G_{lj}$ from $GS_{lx}$

        ● count = count + 1

   g) Set $R = R/count$

   h) Set $AR_{lxk} = R$

  E. Select $Q$ number of most relevant cluster representatives according to $FT_x$ score from $AR_{lx}$ set and form $RSD_{lx}$ sub-data set.

  F. Construct a classifier $C_{lx}$ for $RSD_{lx}$ sub-data set

4. Apply a test sample over all the classifiers of all bootstrapped dataset and calculate the prediction accuracy of each classifier

5. Apply simple voting over all predictions to form an ensemble classifier $EC$ and get final prediction.

6. Calculate number of occurrences for every gene for all $RSD_{lx}$ sub-datasets across all bootstrapped versions and rank them according to their count.

7. Select a number of top-ranked genes as informative genes.

8. End

The block diagram of the proposed MFSAC-EC model is shown in Fig. 2, while the block diagram of the MFSAC method is shown in Fig. 3. The algorithm of the proposed model is described below.

## Description and preprocessing of the datasets

The experimentation has been carried out over ten publicly available different gene expression binary class and multi-class datasets. Among these datasets, eight datasets are cancer datasets and two arthritis datasets. The eight cancer datasets are Leukemia (*Golub et al., 1999*), Colon (*Alon et al., 1999*), Prostate (*Singh et al., 2002*), Lung (*Gordon et al., 2002*), RBreast (*Veer et al., 2002*), Breast (*West et al., 2001*), MLL (*Armstrong et al., 2001*), and SRBCT (*Khan et al., 2001*). To show the accuracy of the proposed model with respect to other than cancer datasets here two arthritis datasets RAHC (*Van der Pouw Kraan et al., 2003*) and RAOA (*van der Pouw Kraan et al., 2007*) are also considered. The summary of the datasets is represented in Table 1.

In the Lesukemia dataset (*Golub et al., 1999*), the gene expression data matrix is prepared using Affymetrix oligonucleotide arrays. The original dataset consists of two datasets: the training dataset and the testing dataset. The training dataset consists of 38 samples (27 Acute Lymphoblastic Leukemia (ALL) and 11 Acute Myeloid Leukemia (AML)) while the test dataset consists of 34 samples (20 Acute Lymphoblastic Leukemia (ALL) and 14 Acute Myeloid Leukemia (AML)), each with 7,129 probes from 6,817 genes. For the Leukemia dataset, training and test datasets are merged here and genes with

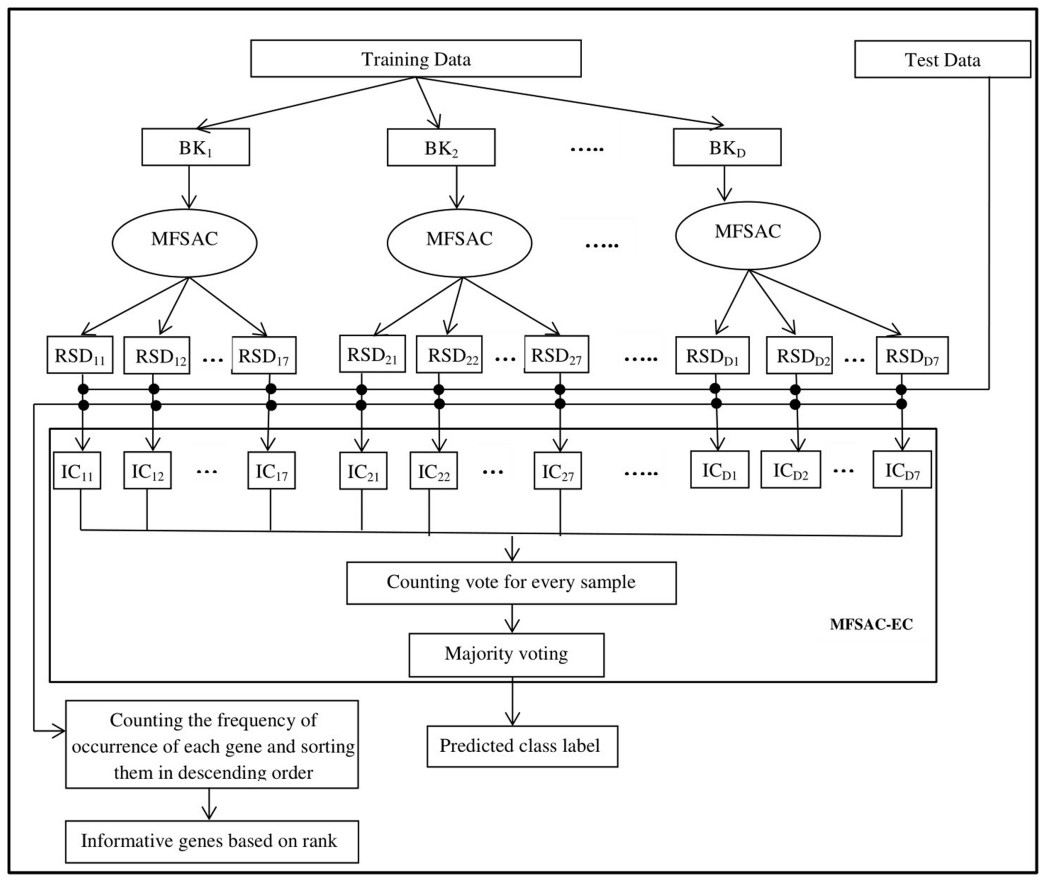

**Figure 2 Block diagram of the proposed MFSAC-EC model.** Here BK$_1$, BK$_2$...BK$_D$ are D number of bootstrapped datasets. RSD$_{11}$...RSD$_{17}$ represent different reduced sub-datasets of BK$_1$ bootstrapped datasets after applying MFSAC method. IC$_{11}$ to IC$_{17}$ represent individual classifiers applied on RSD$_{11}$... RSD$_{17}$ respectively.

missing values are removed and finally, the dataset with 7,070 genes and 72 samples is prepared.

In the Colon cancer dataset (*Alon et al., 1999*), gene expression of 6,500 genes for 62 samples is measured using Affymetrix oligonucleotide arrays. Among these 62 samples, 40 are Colon cancer samples and 22 are normal samples. Among these 6,500 genes, 2,000 genes are selected based on the confidence of measured expression levels.

Prostate cancer dataset (*Singh et al., 2002*) also consists of training and testing datasets. In the training dataset, among 102 samples, 50 are normal samples and 52 are prostate cancer samples. In the test dataset among 34 samples, 25 are prostate cancer samples and 9 are normal prostate samples. Gene expression of every sample is measured with respect to 12,600 genes using Affymetrix chips. Here, training and test datasets are merged, and a dataset with 12,600 genes and 136 samples is formed.

The Lung cancer dataset (*Gordon et al., 2002*) consists of 181 samples. Among these samples, 31 are malignant pleural mesothelioma and rest150 adenocarcinoma of lung cancer. Each sample is represented by 12,533 genes and the gene expression of every sample is measured using Affymetrix human U95A oligonucleotide probe arrays.

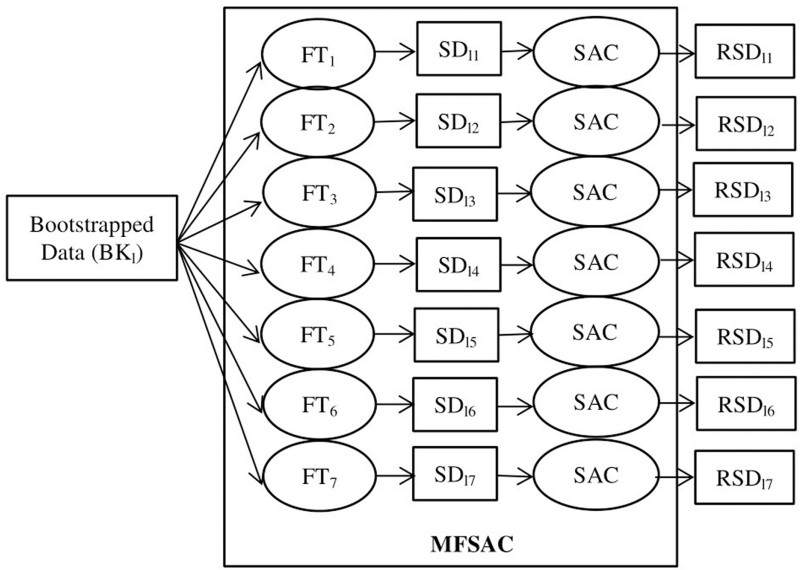

**Figure 3 Block diagram of MFSAC method.** $BK_l$ is the lth bootstrapped dataset. $FT_1...$ $FT_7$ are the seven filter score functions as Table S1. $SD_{11}...SD_{17}$ are sub-datasets created after applying filter score functions. SAC is the Supervised attribute clustering method applied to generate $RSD_{11}...RSD_{17}$ reduced sub-datasets.

**Table 1 Description of cancer gene expression datasets.**

| Dataset | Data Dimension Gene × Sample (Original) | Data Dimension Gene × Sample (Used) | Sample Class Labels | Dataset | Data Dimension Gene × Sample (Original) | Data Dimension Gene × Sample (Used) | Sample Class Labels |
|---|---|---|---|---|---|---|---|
| Leukemia | 7,129 × 72 | 7,070 × 72 | 2 | Breast | 7,129 × 49 | 7,129 × 49 | 2 |
| Colon | 2,000 × 62 | 2,000 × 62 | 2 | MLL | 12,582 × 72 | 12,582 × 72 | 3 |
| Prostate | 12,600 × 136 | 12,600 × 136 | 2 | SRBCT | 2,308 × 63 | 2,308 × 63 | 4 |
| Lung | 12,533 × 181 | 12,533 × 181 | 2 | RAHC | 41,057 × 50 | 41,057 × 50 | 2 |
| Rbreast | 24,481 × 97 | 24,188 × 97 | 2 | RAOA | 18,433 × 30 | 18,433 × 30 | 2 |

In Rbreast data set (*Veer et al., 2002*), the patients, who are considered as breast cancer patients after 5 years intervals of initial diagnosis, fall under the category of relapse and rest as no relapse of metastases. A total of 97 samples have been provided in which 46 patients developed distance metastases within 5 years and they are considered as relapse while the remaining remained healthy and are labeled as non-relapse. This dataset comprises 24,481 genes and among them, 293 are removed.

In the Breast cancer dataset (*West et al., 2001*), the gene expression of 49 samples is measured using HuGeneFL Affymetrix microarray arrays. Breast tumors are positive or negative in the presence or absence of estrogen receptors (ER). In this dataset, 25 samples are ER+ tumors and 24 samples are ER- tumors.

MLL (*Armstrong et al., 2001*) is a type of dataset which comprises of training data set of 57 leukemia samples including 20 ALL, 17 MLL, and 20 AML and the test dataset including four ALL, three MLL, and eight AML samples. For MLL cancer dataset training

and test, datasets are merged here and finally, the dataset with 12,582 genes and 72 samples are prepared.

SRBCT dataset (*Khan et al., 2001*) is introduced as a dataset comprising of gene-expression for identifying small round blue-cell tumors of childhood SRBCT and samples of this dataset are further divided into four class which are neuroblastoma, rhabdomyosarcoma, non-Hodgkin lymphoma, and Ewing family of tumors and they are obtained from cDNA microarrays. A training set consisting of 63 SRBCT tissues, a test set consisting of 20 SRBCT and 5 non-SRBCT samples are available. Here we have considered only the training dataset. Each tissue sample is already standardized to zero mean value and has a unit variance across the genes.

RAHC commonly known as Rheumatoid Arthritis *versus* Healthy Controls is a data set (*Van der Pouw Kraan et al., 2003*) which comprises of gene expression characterizing as peripheral blood cells of 32 patients with RA, three patients with probable RA, and 15 age with sex-matched healthy controls performed under microarrays with a complexity of 26,000 unique genes of 46,000 elements.

RAOA commonly known as Rheumatoid Arthritis *versus* Osteoarthritis is a dataset (*van der Pouw Kraan et al., 2007*) that includes the gene expression of thirty patients in which 21 of them are with RA and the remaining 9 of them are with OA. The Cy5 labeled experimental cDNA and Cy3 labeled common reference sample were pooled and hybridized to the lymphochips (consisting of 18,000 cDNA spots which symbolize immunology in the genes of relevance).

# RESULTS

To assess the performance of the proposed MFSAC-EC model, four well-known existing classifiers named K-Nearest Neighbor (*Duda, Hart & Stork, 1999*), Naive Bayes (*Duda, Hart & Stork, 1999*), Support vector machine (*Vapnik, 1995*), and Decision tree (c4.5) (*Duda, Hart & Stork, 1999*) are applied independently in this model and four different ensemble classification models are formed. To prove the superiority of the proposed model, it is compared with existing well-known filter methods (used here) and existing recognized gene selection methods (*Ding & Peng, 2005*; *Au et al., 2005*; *Maji & Das, 2012*) and also with different existing ensemble classifiers (*Bolón-Canedo, Sánchez-Maroño & Alonso-Betanzos, 2012*; *Nagi & Bhattacharyya, 2013*; *Osareh & Bita, 2013*; *Wang, 2006*; *Wang, Li & Fanget, 2012*). To analyze the performance, the methods are applied to different publicly available cancer and other disease-related gene expression datasets. The major metrics used here for evaluations of the performance of the proposed classifier are the cross-validation method (LOOCV, fivefold, and tenfold), ROC Curve, and Heat map.

## Tools used

The algorithms are implemented using Python programming language and Scikit-learn libraries (*Pedregosa et al., 2011*) which are explained in *Komer, Bergstra & Eliasmith (2014)* for ML algorithms. The programs are executed on an online Colab platform with 12 GB RAM and Intel(R) Xeon(R) processor available in the "CPU" Runtime Type at the time

of writing. Figures and tables are generated in the Matplotlib library (*Hunter, 2007*) and also in Microsoft Excel. The python codes used here are available at https://github.com/NSECResearchCD-SLB/PEERJ_MFSAC_EC.

In the following subsections, first, the different types of metrics used here are discussed, and then the performance of the proposed MFSAC-EC model is verified with respect to these metrics. This is followed by comparing the classification performance of the proposed model with different existing methods in terms of tenfold cross-validation. The proposed model does not only perform the task of classification but also ranks every attribute or gene in descending order based on its information present in the dataset. To show the effectiveness of this ranking procedure topmost eight genes from Colon cancer and Leukemia cancer datasets are represented with their corresponding names, symbols, and references in significant cancer-related journals to demonstrate their significant roles in these cancers.

## Evaluation metrics

The performance of the proposed MFSAC-EC classifier is established with respect to the following measures.

### Cross-validation method

The first well-known metric used here to evaluate the classification model performance is the $k$-fold cross-validation method (*Wang, Li & Fanget, 2012*). In the $k$-fold cross-validation method, the dataset is randomly divided into $k$ number of folds and $k$-1 folds are used for training and one fold is used for testing. The process is repeated for $k$ number of times and average classification accuracy is taken. When $k$ is set at 1 that means the fold size is equal to the size of the dataset (training dataset size is equal to one less than the number of samples in the dataset and validation is done using the remaining sample) then it is considered as Leave one out cross-validation method (LOOCV). For $k$ is equal to two, the cross-validation method is named the household method. It has been found that when $k$ is set at a very small value that means the fold size is large then the accuracy of the classification model is affected by low bias and high variance problems. On the other hand, if $k$ is set at a high value that means the fold size is not so large then the classification accuracy of the classification model has a high bias but low variance. It has been found that tenfold cross-validation method outperforms the LOOCV method (*Breiman & Spector, 1992*; *Ambroise & McLachlan, 2002*; *Asyali et al., 2006*) and it has been also endorsed that the tenfold cross-validation method as a better measure for classification.

In training-testing random splitting the dataset is initially randomly partitioned into training set (2/3$^{rd}$ of the dataset) and testing set (1/3$^{rd}$ of the dataset) with 50 runs.

### ROC curve analysis

The performance of the proposed classifier for two-class datasets is also judged using Receiver Operator Characteristic (ROC) analysis (*Wang, Li & Fanget, 2012*). It is a visual method for evaluating binary classification models. Under this analysis, the following measures are considered to judge the binary classification model.

Classification accuracy ($Acc$) is defined as,

$$Acc = \frac{TP + TN}{TP + FP + TN + FN} \qquad\qquad 0 \leq Acc \leq 1$$

The sensitivity ($SN$) or True Positive Rate ($TPR$) can be defined as,

$$SN = TPR = \frac{TP}{TP + FN}$$

The specificity ($SP$) or True Negative Rate ($TNR$) can be defined as,

$$SP = TNR = \frac{TN}{TN + FP}$$

The False Positive Rate ($FPR$) can be defined as:

$$FPR = (1 - specificity) = \frac{FP}{FP + TN}$$

The Positive Predicted Value ($PPV$) can be defined as:

$$PPV = \frac{TP}{TP + FP}$$

The Negative Predicted Value ($NPV$) can be defined as:

$$NPV = \frac{TN}{TN + FN}$$

where TP, TN, FP, FN are true positive, true negative, false positive, and false negative respectively.

The ROC curve is plotted considering TPR along the y-axis and FPR along the x-axis. The area under the ROC curve (AUC) is used to represent the performance of the binary classification model. The higher AUC value of a ROC curve for a particular classification model signifies the better performance of the classification model in differentiating positive and negative examples. The range of AUC value is 0 <= AUC <= 1.

### Heat map analysis

A heatmap is a data representation diagram in which the values for a variable of interest are portrayed using a data matrix. In this data matrix, the values of the variable are represented across two-axis variables as a grid of colored squares. The axis variables are divided into ranges and each cell's color represents the intensity of that variable for the particular ranges of values of axis variables.

Here, the performance of the proposed classifier for multi-class datasets is judged using Heat map representation of confusion matrix (*Liu et al., 2014*), where a confusion matrix is a tabular representation to visualize the performance of a classification model in terms of true positive, true negative, false positive and false negative.

## Parameter estimation

Before running the MFSAC-EC, the parameters are settled down. In MFSAC-EC the input training dataset is bootstrapped. The proposed MFSAC-EC model is run here varying the number of bootstrapped datasets ($D$) from five to 30 and the classification accuracy of this model is more or less the same from 10 to the rest of the range. So, the number of bootstrapped datasets for every training dataset for this model is set at 10.

In MFSAC method, initially $P$ number of genes is selected by each filter method. Here in Table 2, the classification accuracy of the proposed model is shown with respect to different values of $P$. From Table 2, it has been found that the proposed model gives the best result for $P = 100$ for RAOA and RAHC datasets. In case of Breast cancer, Lung cancer, MLL and SRBCT datasets it gives the best result at $P = 200$. For Leukemia datasets it gives the best result at $P = 500$. So, it can be said that MFSAC-EC gives best result for $P$ value within 200 to 500 in all cases for all datasets except Colon and Prostate. In Colon and Prostate, it shows the best result for $P = 1,500$.

Here we have used SVM, DT (C4.5), NB, and KNN classifiers individually for forming different ensemble classification models. All the classifiers are implemented using Scilit-learn libraries of Python. For all classifiers, we have set parameters with default parameter values. For DT as default setting we have used splitting function = Gini, Splitting criterion = best, height = none (that means for every sample it reaches a leaf/class node). For SVM, we have used the RBF kernel function. For KNN we have chosen K (number of nearest neighbor) value from three to seven.

The overall execution time of a single run of the MFSAC-EC model (considering bootstrapped dataset creation, feature selection using MFSAC, and then generating classification accuracy of test samples using LOOCV, fivefold, tenfold, and random splitting) and testing time using only tenfold are shown for different datasets in Table 3.

## Classification performance of the proposed MFSAC-EC classifier

In Table 4, using the LOOCV method, the classification accuracy of our proposed MFSAC-EC model is 100% for different datasets (Leukemia, Breast, RBreast, Lung, RAOA, and RAHC) for all cases. In the Prostate dataset, we did not get 100% accuracy using our model with respect to any type of existing classifier. In MLL, Colon, and SRBCT it also gives 100% accuracy using all types of ensemble classifiers.

In Tables 5 and 6, it has been shown that using fivefold and tenfold cross-validation, MFSAC-EC does not provide 100% accuracy only for Colon and Prostate cancer datasets. For other datasets, it provides 100% accuracy with respect to all types of ensemble classifiers.

To show the generalization property of the proposed ensemble classifiers, the classification accuracy of these classifiers is also measured repeatedly with respect to the random splitting of the dataset into a training set (2/3 data of original dataset) and test set (1/3 data of original dataset). Random splitting is done with care such that class proportion is alike in the training set and test set. In Table 7, the classification accuracy of the above mentioned four different types of ensemble classifiers for the different number of cluster representatives is shown in different datasets which are based on the best result of

**Table 2 Classification accuracy of MFSAC-EC depending on varying number of genes selected by each filter.** This table shows the impact of parameter P with respect to sample classification accuracy(%) in terms of both LOOCV and tenfold Cross Validation approach. P defines the number of top ranked genes selected by each filter method.

| Dataset | Evaluation Metric | MFSAC-EC | | | | | | | | | | | | | | | | | | | | | | | |
| --- | --- | --- | --- | --- | --- | --- | --- | --- | --- | --- | --- | --- | --- | --- | --- | --- | --- | --- | --- | --- | --- | --- | --- | --- | --- |
| | | P = 100 | | | | P = 200 | | | | P = 500 | | | | P = 1,000 | | | | P = 1,200 | | | | P = 1,500 | | | |
| | | NB | KNN | DT | SVM | NB | KNN | DT | SVM | NB | KNN | DT | SVM | NB | KNN | DT | SVM | NB | KNN | DT | SVM | NB | KNN | DT | SVM |
| Leukemia | LOOCV | 98.6 | 98.6 | 98.6 | 98.6 | 98.6 | 98.6 | 98.6 | 98.6 | 100 | 100 | 100 | 100 | 100 | 100 | 100 | 100 | 100 | 100 | 100 | 100 | 100 | 100 | 100 | 100 |
| | 10 Fold | 98.6 | 98.6 | 98.6 | 98.6 | 98.6 | 98.6 | 98.6 | 98.6 | 100 | 100 | 100 | 100 | 100 | 100 | 100 | 100 | 100 | 100 | 100 | 100 | 100 | 100 | 100 | 100 |
| RAHC | LOOCV | 100 | 100 | 100 | 100 | 100 | 100 | 100 | 100 | 100 | 100 | 100 | 100 | 100 | 100 | 100 | 100 | 100 | 100 | 100 | 100 | 100 | 100 | 100 | 100 |
| | 10 Fold | 100 | 100 | 100 | 100 | 100 | 100 | 100 | 100 | 100 | 100 | 100 | 100 | 100 | 100 | 100 | 100 | 100 | 100 | 100 | 100 | 100 | 100 | 100 | 100 |
| MLL | LOOCV | 98.6 | 100 | 100 | 97.2 | 100 | 100 | 100 | 100 | 100 | 100 | 100 | 100 | 100 | 100 | 100 | 100 | 100 | 100 | 100 | 100 | 100 | 100 | 100 | 100 |
| | 10 Fold | 97.2 | 100 | 100 | 97.2 | 100 | 100 | 100 | 100 | 100 | 100 | 100 | 100 | 100 | 100 | 100 | 100 | 100 | 100 | 100 | 100 | 100 | 100 | 100 | 100 |
| RAOA | LOOCV | 100 | 100 | 100 | 100 | 100 | 100 | 100 | 100 | 100 | 100 | 100 | 100 | 100 | 100 | 100 | 100 | 100 | 100 | 100 | 100 | 100 | 100 | 100 | 100 |
| | 10 Fold | 100 | 100 | 100 | 100 | 100 | 100 | 100 | 100 | 100 | 100 | 100 | 100 | 100 | 100 | 100 | 100 | 100 | 100 | 100 | 100 | 100 | 100 | 100 | 100 |
| SRBCT | LOOCV | 98.4 | 98.4 | 100 | 98.4 | 100 | 100 | 100 | 100 | 100 | 100 | 100 | 100 | 100 | 100 | 100 | 100 | 100 | 100 | 100 | 100 | 100 | 100 | 100 | 100 |
| | 10 Fold | 100 | 98.4 | 98.4 | 98.4 | 100 | 100 | 100 | 100 | 100 | 100 | 100 | 100 | 100 | 100 | 100 | 100 | 100 | 100 | 100 | 100 | 100 | 100 | 100 | 100 |
| Breast | LOOCV | 98 | 95.9 | 93.9 | 95.9 | 100 | 100 | 100 | 100 | 100 | 100 | 100 | 100 | 100 | 100 | 100 | 100 | 100 | 100 | 100 | 100 | 100 | 100 | 100 | 100 |
| | 10 Fold | 100 | 95.9 | 95.9 | 98 | 100 | 100 | 100 | 100 | 100 | 100 | 100 | 100 | 100 | 100 | 100 | 100 | 100 | 100 | 100 | 100 | 100 | 100 | 100 | 100 |
| Lung | LOOCV | 100 | 100 | 100 | 100 | 100 | 100 | 100 | 100 | 100 | 100 | 100 | 100 | 100 | 100 | 100 | 100 | 100 | 100 | 100 | 100 | 100 | 100 | 100 | 100 |
| | 10 Fold | 100 | 100 | 100 | 100 | 100 | 100 | 100 | 100 | 100 | 100 | 100 | 100 | 100 | 100 | 100 | 100 | 100 | 100 | 100 | 100 | 100 | 100 | 100 | 100 |
| Rbreast | LOOCV | 92.6 | 93.7 | 93.7 | 95.8 | 99 | 97.9 | 100 | 99 | 100 | 100 | 100 | 100 | 100 | 100 | 100 | 100 | 100 | 100 | 100 | 100 | 100 | 100 | 100 | 100 |
| | 10 Fold | 91.6 | 96.8 | 95.8 | 97.9 | 97.9 | 99 | 99 | 99 | 100 | 100 | 100 | 100 | 100 | 100 | 100 | 100 | 100 | 100 | 100 | 100 | 100 | 100 | 100 | 100 |
| COLON | LOOCV | 91.9 | 91.9 | 93.6 | 91.9 | 91.9 | 91.9 | 96.8 | 91.9 | 98.4 | 98.4 | 96.8 | 98.4 | 98.4 | 98.4 | 98.4 | 98.4 | 98.4 | 98.4 | 98.4 | 100 | 100 | 100 | 98.4 | 100 |
| | 10 Fold | 91.9 | 91.9 | 93.6 | 91.9 | 91.9 | 91.9 | 95.2 | 91.9 | 98.4 | 96.8 | 96.8 | 96.8 | 98.4 | 100 | 98.4 | 98.4 | 100 | 98.4 | 98.4 | 100 | 100 | 100 | 98.4 | 100 |
| Prostrate | LOOCV | 83.8 | 90.4 | 92.7 | 86.8 | 88.2 | 92.7 | 92.7 | 88.2 | 91.2 | 95.6 | 97.1 | 91.9 | 94.9 | 97.1 | 97.8 | 94.1 | 98.5 | 98.5 | 97.8 | 98.5 | 98.5 | 99.3 | 98.5 | 99.3 |
| | 10 Fold | 85.3 | 88.2 | 91.9 | 86.8 | 88.2 | 91.9 | 93.4 | 89.7 | 91.9 | 95.6 | 97.8 | 91.2 | 95.6 | 97.8 | 97.8 | 94.1 | 99.3 | 98.5 | 97.8 | 98.5 | 99.3 | 99.3 | 97.8 | 99.3 |

**Table 3 Total execution time in a single run of MFSAC-EC on different datasets.** Total execution time in a single run of MFSAC-EC including Bootstrapped dataset creation, Feature Selection by filter methods and supervised attribute clustering approach, Training, Testing using LOOCV, fivefold, tenfold, and Random Splitting is given in the first row. While execution time using only tenfold Cross Validation is given in the 2nd row. Here the time for the best $P$ value is shown here.

|  | Leukemia | RAHC | MLL | RAOA | SRBCT | Breast | Lung | Rbreast | COLON | Prostrate |
|---|---|---|---|---|---|---|---|---|---|---|
| No. of Feature selected for best result | 500 | 100 | 200 | 100 | 200 | 200 | 100 | 500 | 1,200 | 3,000 |
| Total Time Taken | 8 min 23 s | 7 min 32 s | 7 min 54 s | 4 min 43 s | 5 min 17 s | 4 min 2 s | 11 min 14 s | 10 min 22 s | 17 min 40 s | 1 h 18 min 41 s |
| Time Taken for only 10 fold | 35 s | 30 s | 41 s | 36 s | 30 s | 32 s | 36 s | 33 s | 30 s | 36 s |

**Table 4 Classification accuracy of the proposed MFSAC-EC model with respect to LOOCV.** Classification accuracy (%) of MFSAC-EC model has been shown in terms of LOOCV with respect to four ensemble classifiers MFSAC-EC + NB, MFSAC-EC+KNN, MFSAC-EC+DT, and MFSAC-EC+SVM. Every ensemble classifier is run 50 times using LOOCV for every dataset and the accuracy is shown which is obtained maximum number of times.

| Dataset | Proposed model | | Cluster representatives | | | Dataset | Proposed model | | Cluster representatives | | |
|---|---|---|---|---|---|---|---|---|---|---|---|
| | | | 1 | 2 | 3 | | | | 1 | 2 | 3 |
| COLON | MFSAC-EC | NB | 100 | 98.39 | 98.39 | MLL | MFSAC-EC | NB | 100 | 100 | 100 |
| | | KNN | 98.39 | 100 | 100 | | | KNN | 100 | 100 | 100 |
| | | DT | 98.39 | 98.39 | 98.39 | | | DT | 100 | 100 | 100 |
| | | SVM | 100 | 98.4 | 98.4 | | | SVM | 100 | 100 | 100 |
| Prostate | | NB | 97.06 | 97.79 | 98.53 | SRBCT | | NB | 96.83 | 100 | 100 |
| | | KNN | 97.79 | 97.79 | 98.53 | | | KNN | 96.83 | 100 | 100 |
| | | DT | 97.79 | 98.53 | 97.79 | | | DT | 96.83 | 98.41 | 100 |
| | | SVM | 98.53 | 99.26 | 99.26 | | | SVM | 82.54 | 98.41 | 100 |
| Leukemia | | NB | 100 | 100 | 100 | Lung | | NB | 100 | 100 | 100 |
| | | KNN | 100 | 100 | 100 | | | KNN | 100 | 100 | 100 |
| | | DT | 100 | 100 | 100 | | | DT | 100 | 100 | 100 |
| | | SVM | 100 | 100 | 100 | | | SVM | 100 | 100 | 100 |
| RAOA | | NB | 100 | 100 | 100 | RAHC | | NB | 100 | 100 | 100 |
| | | KNN | 100 | 100 | 100 | | | KNN | 100 | 100 | 100 |
| | | DT | 100 | 100 | 100 | | | DT | 100 | 100 | 100 |
| | | SVM | 100 | 100 | 100 | | | SVM | 100 | 100 | 100 |
| Breast | | NB | 100 | 100 | 100 | RBreast | | NB | 100 | 100 | 100 |
| | | KNN | 100 | 100 | 100 | | | KNN | 100 | 100 | 100 |
| | | DT | 100 | 100 | 100 | | | DT | 100 | 100 | 100 |
| | | SVM | 100 | 100 | 100 | | | SVM | 100 | 100 | 100 |

50 random splitting of the dataset into a training set (2/3 data of original dataset) and test set (1/3 data of original dataset).

From the results of Tables 4 to 7, it has been observed that classification accuracy in the LOOCV method, fivefold cross-validation, and tenfold cross-validation methods is

**Table 5 Classification accuracy of the proposed MFSAC-EC model with respect to fivefold cross validation.** Classification accuracy (%) of MFSAC-EC model has been shown in terms of fivefold Cross Validation with respect to four ensemble classifiers MFSAC-EC + NB, MFSAC-EC +KNN, MFSAC-EC+DT, and MFSAC-EC+SVM. Every ensemble classifier is run 50 times using fivefold Cross Validation for every dataset and the accuracy is shown which is obtained maximum number of times.

| Dataset | Proposed model | | Cluster representatives | | | Dataset | Proposed model | | Cluster representatives | | |
|---|---|---|---|---|---|---|---|---|---|---|---|
| | | | 1 | 2 | 3 | | | | 1 | 2 | 3 |
| COLON | MFSAC-EC | NB | 96.77 | 96.77 | 96.77 | MLL | MFSAC-EC | NB | 100 | 100 | 100 |
| | | KNN | 98.39 | 96.77 | 96.77 | | | KNN | 98.61 | 100 | 100 |
| | | DT | 98.39 | 96.77 | 98.39 | | | DT | 98.61 | 100 | 100 |
| | | SVM | 98.39 | 96.77 | 96.77 | | | SVM | 100 | 100 | 100 |
| Prostate | | NB | 97.06 | 97.79 | 98.53 | SRBCT | | NB | 98.41 | 100 | 100 |
| | | KNN | 97.79 | 97.79 | 99.26 | | | KNN | 96.83 | 100 | 100 |
| | | DT | 97.06 | 97.79 | 94.85 | | | DT | 96.83 | 98.41 | 100 |
| | | SVM | 97.79 | 98.53 | 99.26 | | | SVM | 96.83 | 100 | 100 |
| Leukemia | | NB | 100 | 100 | 100 | Lung | | NB | 100 | 100 | 100 |
| | | KNN | 100 | 100 | 100 | | | KNN | 100 | 100 | 100 |
| | | DT | 100 | 100 | 100 | | | DT | 100 | 99.44 | 100 |
| | | SVM | 100 | 100 | 100 | | | SVM | 100 | 100 | 100 |
| RAOA | | NB | 100 | 100 | 100 | RAHC | | NB | 100 | 100 | 100 |
| | | KNN | 100 | 100 | 100 | | | KNN | 100 | 100 | 100 |
| | | DT | 100 | 100 | 100 | | | DT | 100 | 100 | 100 |
| | | SVM | 100 | 100 | 100 | | | SVM | 100 | 100 | 100 |
| Breast | | NB | 100 | 100 | 100 | RBreast | | NB | 100 | 100 | 100 |
| | | KNN | 100 | 100 | 100 | | | KNN | 100 | 100 | 100 |
| | | DT | 100 | 100 | 100 | | | DT | 100 | 100 | 100 |
| | | SVM | 100 | 100 | 100 | | | SVM | 100 | 100 | 100 |

higher than the random splitting of the dataset, and the overall generalization performance of the proposed classification model is also good.

The performance of the proposed model for different two-class datasets with respect to different parameters like SN, SP, PPV, NPV, FPR is shown in Table 8. From this table, it is found that the performance of the proposed model is very good with respect to all these parameters for all two-class datasets.

In Fig. 4, the ROC curve is shown for different two-class datasets. In Figs. 4A, 4B, and 4C, the ROC curves are shown for Breast cancer using LOOCV, for Colon cancer using fivefold cross validation, and for RAHC dataset using tenfold cross-validation respectively. The ROC curves for Leukemia Cancer, and Lung cancer datasets using LOOCV are given in Figs. S1A and S1B respectively. For Breast cancer, Leukemia cancer, and Lung cancer, the AUC value is equal to 1.0 in every case. The ROC curves are shown for RAOA, and RBreast cancer datasets using fivefold cross-validation in Figs. S2A, and S2B respectively. For these datasets also the prediction accuracy using fivefold cross validation is very high according to the AUC value. In Fig. S2C, the ROC curves are shown

**Table 6 Classification accuracy of the proposed MFSAC-EC model with respect to tenfold cross validation.** Classification accuracy (%) of MFSAC-EC model has been shown in terms of tenfold cross validation with respect to four ensemble classifiers MFSAC-EC + NB, MFSAC-EC +KNN, MFSAC-EC+DT, and MFSAC-EC+SVM. Every ensemble classifier is run 50 times using tenfold cross validation for every dataset and the accuracy is shown which is obtained maximum number of times.

| Dataset | Proposed model | | Cluster representatives | | | Dataset | Proposed model | | Cluster representatives | | |
|---|---|---|---|---|---|---|---|---|---|---|---|
| | | | 1 | 2 | 3 | | | | 1 | 2 | 3 |
| COLON | MFSAC-EC | NB | 98.39 | 98.39 | 98.39 | MLL | MFSAC-EC | NB | 100 | 100 | 100 |
| | | KNN | 98.39 | 98.39 | 100 | | | KNN | 100 | 100 | 100 |
| | | DT | 98.39 | 98.39 | 98.39 | | | DT | 100 | 100 | 100 |
| | | SVM | 98.39 | 98.39 | 98.39 | | | SVM | 100 | 100 | 100 |
| Prostate | | NB | 97.06 | 97.79 | 98.53 | SRBCT | | NB | 96.83 | 96.83 | 100 |
| | | KNN | 97.79 | 97.79 | 99.26 | | | KNN | 92.06 | 100 | 100 |
| | | DT | 97.06 | 97.79 | 94.85 | | | DT | 95.24 | 96.83 | 100 |
| | | SVM | 97.79 | 98.53 | 99.26 | | | SVM | 80.95 | 92.06 | 100 |
| Leukemia | | NB | 100 | 100 | 100 | Lung | | NB | 100 | 100 | 100 |
| | | KNN | 100 | 100 | 100 | | | KNN | 100 | 100 | 100 |
| | | DT | 100 | 100 | 100 | | | DT | 100 | 100 | 100 |
| | | SVM | 100 | 100 | 100 | | | SVM | 100 | 100 | 100 |
| Breast | | NB | 100 | 100 | 100 | RBreast | | NB | 100 | 100 | 100 |
| | | KNN | 100 | 100 | 100 | | | KNN | 100 | 100 | 100 |
| | | DT | 100 | 100 | 100 | | | DT | 100 | 100 | 100 |
| | | SVM | 100 | 100 | 100 | | | SVM | 100 | 100 | 100 |
| RAOA | | NB | 100 | 100 | 100 | RAHC | | NB | 100 | 100 | 100 |
| | | KNN | 100 | 100 | 100 | | | KNN | 100 | 100 | 100 |
| | | DT | 100 | 100 | 100 | | | DT | 100 | 100 | 100 |
| | | SVM | 100 | 100 | 100 | | | SVM | 100 | 100 | 100 |

for Prostate cancer using tenfold cross-validation. From these curves of tenfold cross validation, it may be seen that except for Prostate cancer, for all other datasets the AUC value is 1 and for Prostate cancer, the AUC value is close to 1.

In Figs. 5A and 5B, heatmap representation of the confusion matrix are shown for multi-class datasets: SRBCT and MLL with respect to fivefold cross-validation, and tenfold cross-validation respectively. From these figures, it is clear that for the proposed model prediction accuracy is accurate in most cases.

## Comparison of MFSAC-EC model with well-known existing filter methods used in this model

In Fig. S3, the proposed MFSAC-EC model in combination with different existing classifiers is compared with different filter methods used in this model with respect to SRBCT, RAHC, Prostate, and Colon datasets in terms of tenfold cross-validation. In all cases, the performance of the proposed model is significantly better with respect to all filters.

**Table 7 Classification accuracy of the proposed MFSAC-EC model with respect to random splitting of the datasets.** Classification accuracy (%) of MFSAC-EC model has been shown in terms of random splitting with respect to four ensemble classifiers MFSAC-EC + NB, MFSAC-EC+KNN, MFSAC-EC+DT, and MFSAC-EC+SVM. Every ensemble classifier is run 50 times using random splitting for every dataset and the accuracy is shown which is obtained maximum number of times. For random splitting the dataset is divided into training (2/3) and testing (1/3) part 50 times randomly.

| Dataset | Proposed model | | Cluster representatives | | | Dataset | Proposed model | | Cluster representatives | | |
|---|---|---|---|---|---|---|---|---|---|---|---|
| | | | 1 | 2 | 3 | | | | 1 | 2 | 3 |
| COLON | MFSAC-EC | NB | 98.39 | 98.39 | 98.39 | MLL | MFSAC-EC | NB | 100 | 100 | 100 |
| | | KNN | 98.39 | 98.39 | 98.39 | | | KNN | 100 | 100 | 100 |
| | | DT | 98.39 | 98.39 | 98.39 | | | DT | 98.61 | 100 | 98.61 |
| | | SVM | 98.39 | 100 | 98.39 | | | SVM | 100 | 100 | 100 |
| Prostate | | NB | 94.68 | 95.74 | 93.62 | SRBCT | | NB | 95 | 85 | 95 |
| | | KNN | 97.87 | 96.81 | 92.55 | | | KNN | 95 | 100 | 90 |
| | | DT | 94.68 | 94.68 | 94.68 | | | DT | 80 | 90 | 95 |
| | | SVM | 94.68 | 96.81 | 94.68 | | | SVM | 65 | 75 | 95 |
| Leukemia | | NB | 100 | 100 | 100 | Lung | | NB | 100 | 100 | 100 |
| | | KNN | 100 | 100 | 100 | | | KNN | 100 | 100 | 100 |
| | | DT | 100 | 100 | 100 | | | DT | 100 | 100 | 100 |
| | | SVM | 100 | 100 | 100 | | | SVM | 100 | 100 | 100 |
| RAOA | | NB | 100 | 100 | 100 | RAHC | | NB | 100 | 100 | 100 |
| | | KNN | 100 | 100 | 100 | | | KNN | 100 | 100 | 100 |
| | | DT | 100 | 100 | 100 | | | DT | 100 | 100 | 81.25 |
| | | SVM | 100 | 100 | 100 | | | SVM | 100 | 100 | 81.25 |
| Breast | | NB | 100 | 100 | 100 | RBreast | | NB | 91.94 | 91.94 | 91.94 |
| | | KNN | 100 | 100 | 100 | | | KNN | 85.48 | 87.10 | 83.87 |
| | | DT | 100 | 100 | 100 | | | DT | 83.87 | 79.03 | 80.65 |
| | | SVM | 100 | 100 | 100 | | | SVM | 93.55 | 91.94 | 91.94 |

## Comparison of MFSAC-EC Model with Well-Known Existing Gene Selection Methods

In Fig. 6, the MFSAC-EC model with different existing classifiers as base classifiers are compared with existing well-known supervised gene selection methods named mRMR (minimum redundancy maximum relevance framework) (*Ding & Peng, 2005*), MSG (mutual information based supervised gene clustering algorithm) (*Maji & Das, 2012*), CFS (Correlation-based Feature Selection) (*Ruiz, Riquelme & Aguilar-Ruiz, 2006*), and FCBF (Fast Correlation-Based Filter) (*Ruiz, Riquelme & Aguilar-Ruiz, 2006*) with respect to different classifiers using tenfold cross-validation method. From these results, it has been found that the proposed model outperforms in most of the cases.

In Fig. 7, the MFSAC-EC model is compared with well-known existing unsupervised gene selection methods named MGSACO (*Tabakhi et al., 2015*), UFSACO (*Tabakhi, Moradi & Akhlaghian, 2014*), RSM (*Lai, Reinders & Wessels, 2006*), MC (*Haindl et al., 2006*), RRFS (*Ferreira & Figueiredo, 2012*), TV (*Theodoridis & Koutroumbas, 2008*), and LS (*Liao et al., 2014*) with respect to DT, SVM, NB classifiers using random splitting

**Table 8 Evaluation of MFSAC-EC classifier based on SN, SP, PPV, NPV, FPR for two class data sets with respect to LOOCV.** The performance of the MFSAC-EC model for two class datasets is represented using Receiver Operator Characteristic (ROC) analysis. SN represents Sensitivity, SP represents Specificity, PPV represents Positive Predicted Value, NPV represents Negative Predicted Value, and FPR represents False Positive Rate.

| Dataset | Proposed model | | SN | SP | PPV | NPV | FPR | Dataset | Proposed model | | SN | SP | PPV | NPV | FPR |
|---|---|---|---|---|---|---|---|---|---|---|---|---|---|---|---|
| Leukemia | MFSGC-EC | NB | 100 | 100 | 100 | 100 | 0 | Breast | MFSGC-EC | NB | 100 | 100 | 100 | 100 | 0 |
| | | KNN | 100 | 100 | 100 | 100 | 0 | | | KNN | 100 | 100 | 100 | 100 | 0 |
| | | DT | 100 | 100 | 100 | 100 | 0 | | | DT | 100 | 100 | 100 | 100 | 0 |
| | | SVM | 100 | 100 | 100 | 100 | 0 | | | SVM | 100 | 100 | 100 | 100 | 0 |
| Prostate | | NB | 98.7 | 98.3 | 98.7 | 98.3 | 1.7 | Rbreast | | NB | 100 | 100 | 100 | 100 | 0 |
| | | KNN | 98.7 | 98.3 | 98.7 | 98.3 | 1.7 | | | KNN | 100 | 100 | 100 | 100 | 0 |
| | | DT | 100 | 96.61 | 97.46 | 100 | 3.4 | | | DT | 100 | 100 | 100 | 100 | 0 |
| | | SVM | 100 | 98.3 | 98.7 | 100 | 1.7 | | | SVM | 100 | 100 | 100 | 100 | 0 |
| Colon | | NB | 100 | 100 | 100 | 100 | 0 | Lung | | NB | 100 | 100 | 100 | 100 | 0 |
| | | KNN | 100 | 100 | 100 | 100 | 0 | | | KNN | 100 | 100 | 100 | 100 | 0 |
| | | DT | 100 | 100 | 100 | 100 | 0 | | | DT | 100 | 100 | 100 | 100 | 0 |
| | | SVM | 100 | 100 | 100 | 100 | 0 | | | SVM | 100 | 100 | 100 | 100 | 0 |
| RAHC | | NB | 100 | 100 | 100 | 100 | 0 | RAOA | | NB | 100 | 100 | 100 | 100 | 0 |
| | | KNN | 100 | 100 | 100 | 100 | 0 | | | KNN | 100 | 100 | 100 | 100 | 0 |
| | | DT | 100 | 100 | 100 | 100 | 0 | | | DT | 100 | 100 | 100 | 100 | 0 |
| | | SVM | 100 | 100 | 100 | 100 | 0 | | | SVM | 100 | 100 | 100 | 100 | 0 |

method. From these results, it can be said that the MFSAC-EC model outperforms in all cases.

## Comparison of MFSAC-EC model with well-known existing ensemble classification and DEEP learning models

In Table 9, the proposed MFSAC-EC model using the DT classifier is compared with well-known existing ensemble classification models with respect to tenfold cross-validation. These models are PCA-basedRotBoost (*Osareh & Bita, 2013*), ICA-based RotBoost (*Osareh & Bita, 2013*), AdaBoost (*Osareh & Bita, 2013*), Bagging (*Osareh & Bita, 2013*), Arcing (*Osareh & Bita, 2013*), Rotation Forest (*Osareh & Bita, 2013*), EN-NEW1 (*Wang, 2006*), and EN-NEW2 (*Wang, 2006*). From Table 9, it is clear that the proposed model using DT classifier outperforms in all cases.

In Table 10, the proposed MFSAC-EC model using DT, NB, KNN as base classifiers are compared with different existing ensemble classifiers with respect to tenfold cross-validation. These classifiers are Bagging based ensemble classifier (*Nagi & Bhattacharyya, 2013*), Boosting based ensemble classifier (*Nagi & Bhattacharyya, 2013*), Stacking based ensemble classifier (*Nagi & Bhattacharyya, 2013*), Heuristic breadth-first search-based ensemble classifier (HBSA) (*Wang, Li & Fanget, 2012*), Sd_Ens (*Nagi & Bhattacharyya, 2013*), and Meta_Ens (*Nagi & Bhattacharyya, 2013*). In Table 11 our model using SVM and KNN as base classifiers is compared with auto-encoder-based deep learning models (*Nabendu, Pintu & Pratyay, 2020*) in terms of random splitting. Here, results are shown only for the datasets for which results are available in the literature, and all other

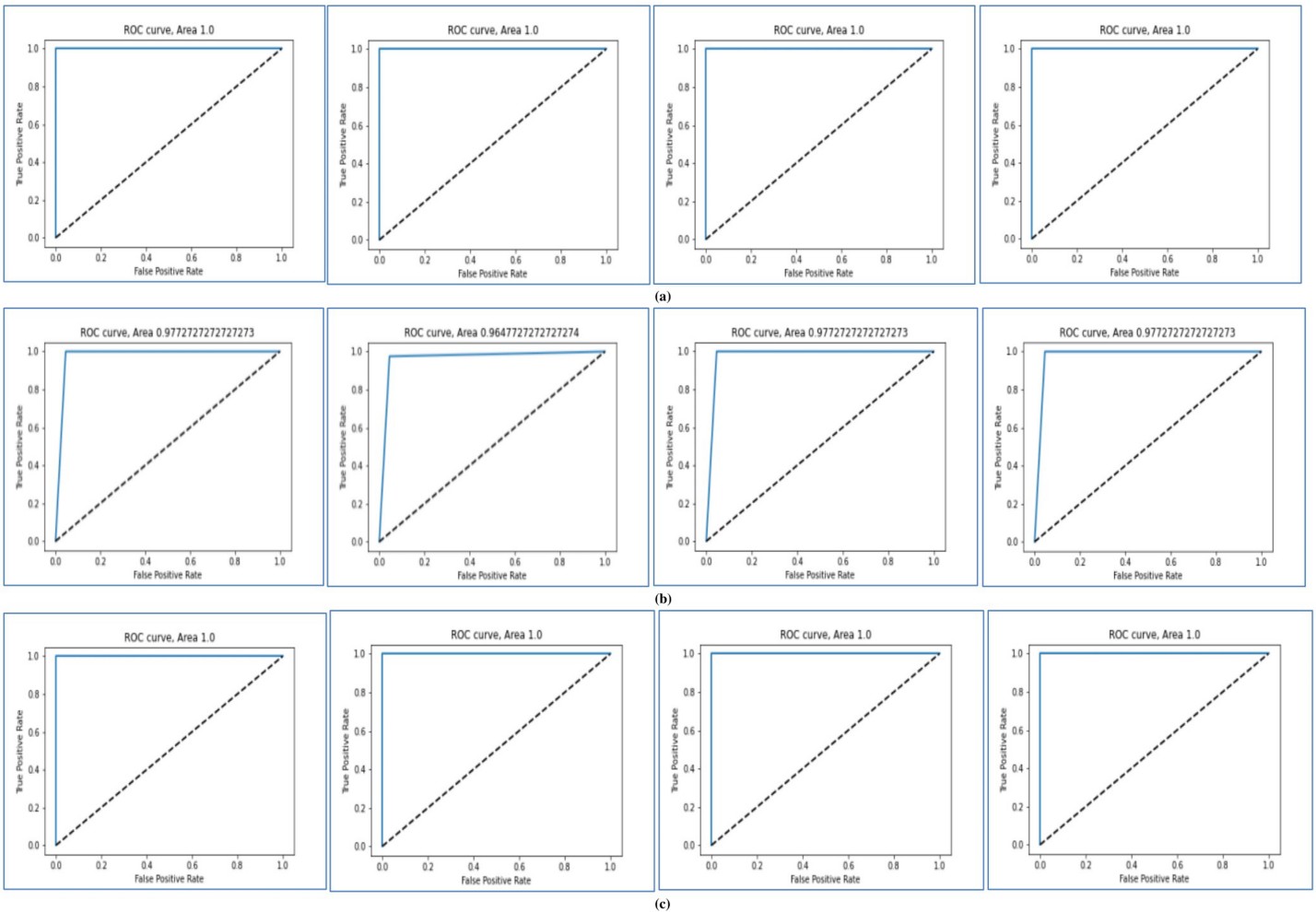

**Figure 4 AUC for for three datasets using MFSAC-EC+ KNN, MFSAC-EC+NB, MFSAC-EC+ DT and MFSAC-EC+ SVM classifiers.** (A) For the breast cancer dataset using LOOCV. (B) For the colon cancer dataset using fivefold cross validation. (C) For RAHC dataset using tenfold cross validation.

fields are marked as "Not Found". In all cases, the MFSAC-EC model outperforms all the well-known existing ensemble models (except for the Colon cancer dataset) and deep learning models which in turn validates the usefulness of the proposed model.

## Biological significance analysis

The top eight genes selected by the MFSAC-EC model for Colon cancer and Leukemia are listed in Table 12. For every gene, the name and symbol of the gene as well as the Accession number of the Affymetrix chip are listed. Apart from this information, to validate those genes, biomedical literature of the genes is searched and for every gene, the corresponding reference about its role and significance for a particular disease is provided.

## DISCUSSION

In this paper, a new Multiple Filtering and Supervised Attribute Clustering algorithm-based ensemble classification model named MFSAC-EC is proposed. The main motivation

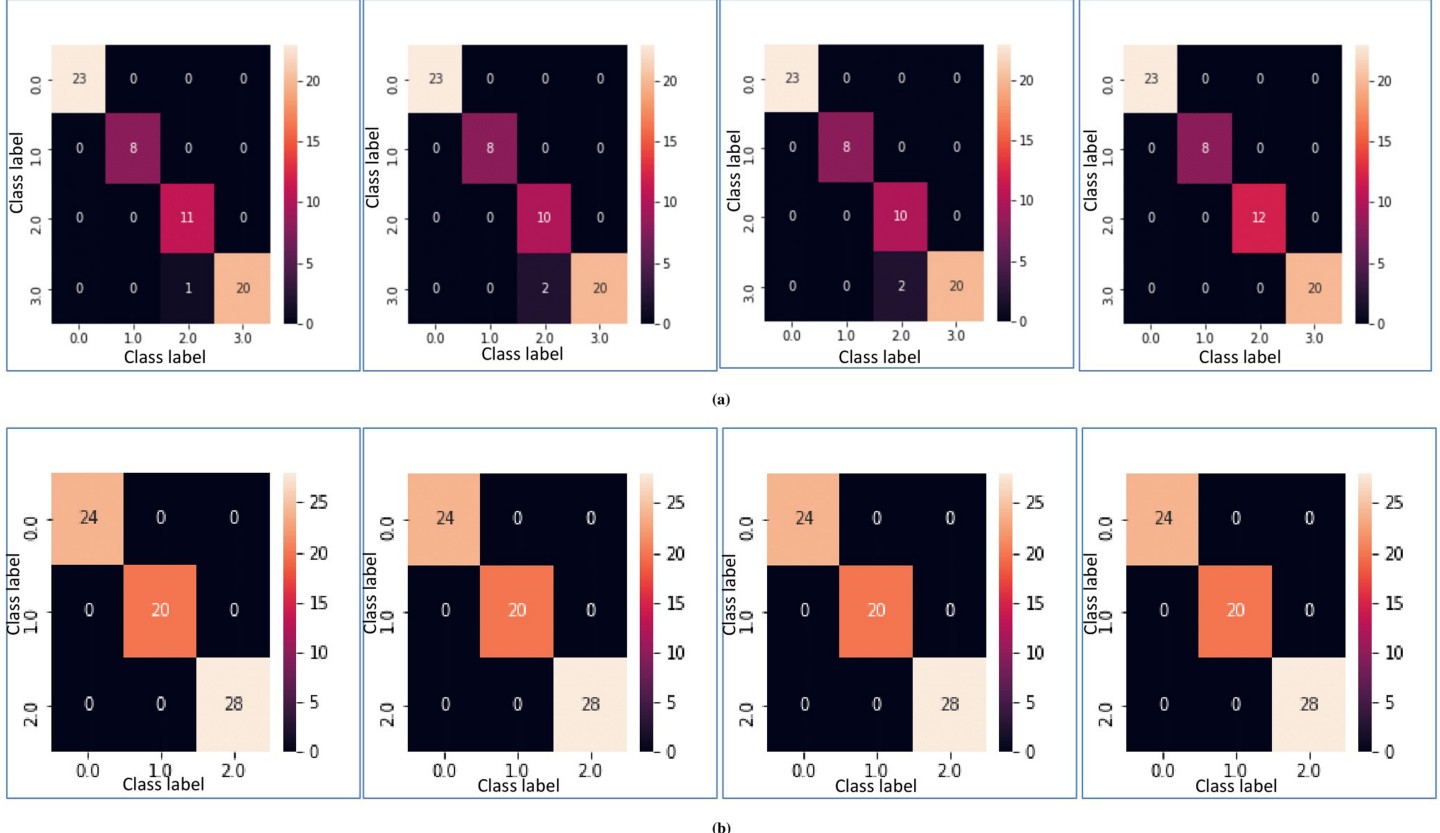

**Figure 5 Heatmap of MFSAC-EC with base classifiers NB, KNN, DT and SVM, respectively, for multiclass datasets.** (A) For the SRBCT dataset using fivefold cross validation. (B) For MLL dataset using tenfold cross validation.

behind this work is to develop a machine learning-based ensemble classification model to overcome the over-fitting problem which arises due to the presence of sample class imbalance problem, small sample size problem, and also high dimensional feature set problem in the microarray gene expression dataset, to enhance the prediction capability of the proposed model.

Nowadays, in designing machine learning models, the use of ensemble methodology has been increasing day by day as it incorporates multiple learning algorithms and also training datasets in different efficient manners to improve the overall prediction accuracy of the model. Due to the inclusion of prediction accuracy of multiple learning models and also the use of different bootstrapping datasets, the chances of potential overfitting in training data is greatly reduced in the ensemble models, and as a consequence the prediction accuracy increases. One necessary condition of the superior performance of an ensemble classifier with respect to its individual member/base classifier is that every base classifier should be very accurate and diverse (*Osareh & Bita, 2013*). A classifier is considered accurate if its generalization capability is high and two classifiers satisfy diverse property if their prediction in classifying the same unknown samples vary from each other. The general principle of ensemble methods is to rearrange training datasets in different

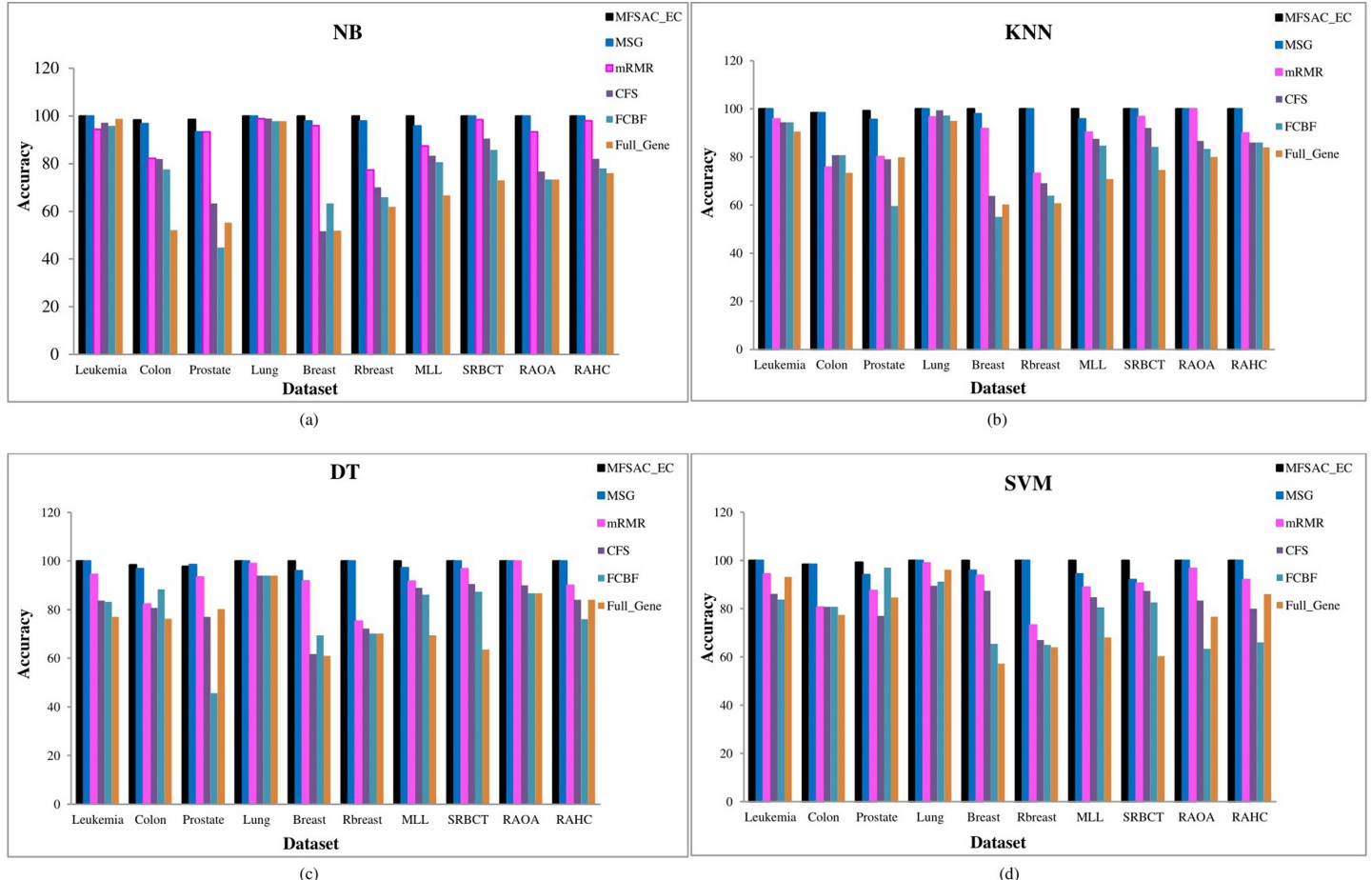

**Figure 6 Comparison of MFSAC-EC with other well-known supervised gene selection methods and full gene set in terms of fivefold cross validation for all datasets.** In each figure classification accuracy (%) of MFSAC-EC model along with other supervised gene selection methods for all datasets are represented using different colored bars using (A) NB (B) KNN (c) DT and (D) SVM as base classifier.

ways (either by resampling or reweighting) and build an ensemble of base classifiers by applying a base classifier on every rearranged training dataset (*Osareh & Bita, 2013*).

In our proposed ensemble model, at first, a number of bootstrapped datasets of the original training dataset is created. In every bootstrapped dataset, the class imbalance problem is solved using the oversampling method. Then for every bootstrapped dataset, a number of sub-datasets are created using the MFSAC method (which is a hybrid method combining multi-filters and a new supervised attribute/gene clustering method), and then for every generated sub-dataset, a base classifier is constructed using any existing classification model. After that, a new ensemble classifier (EC) is formed using the majority voting scheme by combining the prediction accuracy of all those base classifiers.

The prediction accuracy of the proposed model is verified by applying it to high-dimensional microarray gene expression data From Figs. 6, and 7 it has been found that the classification accuracy of the MFSAC-EC model is much better than the well-known existing gene selection methods. From Tables 9, 10, and 11, it has been also found that the

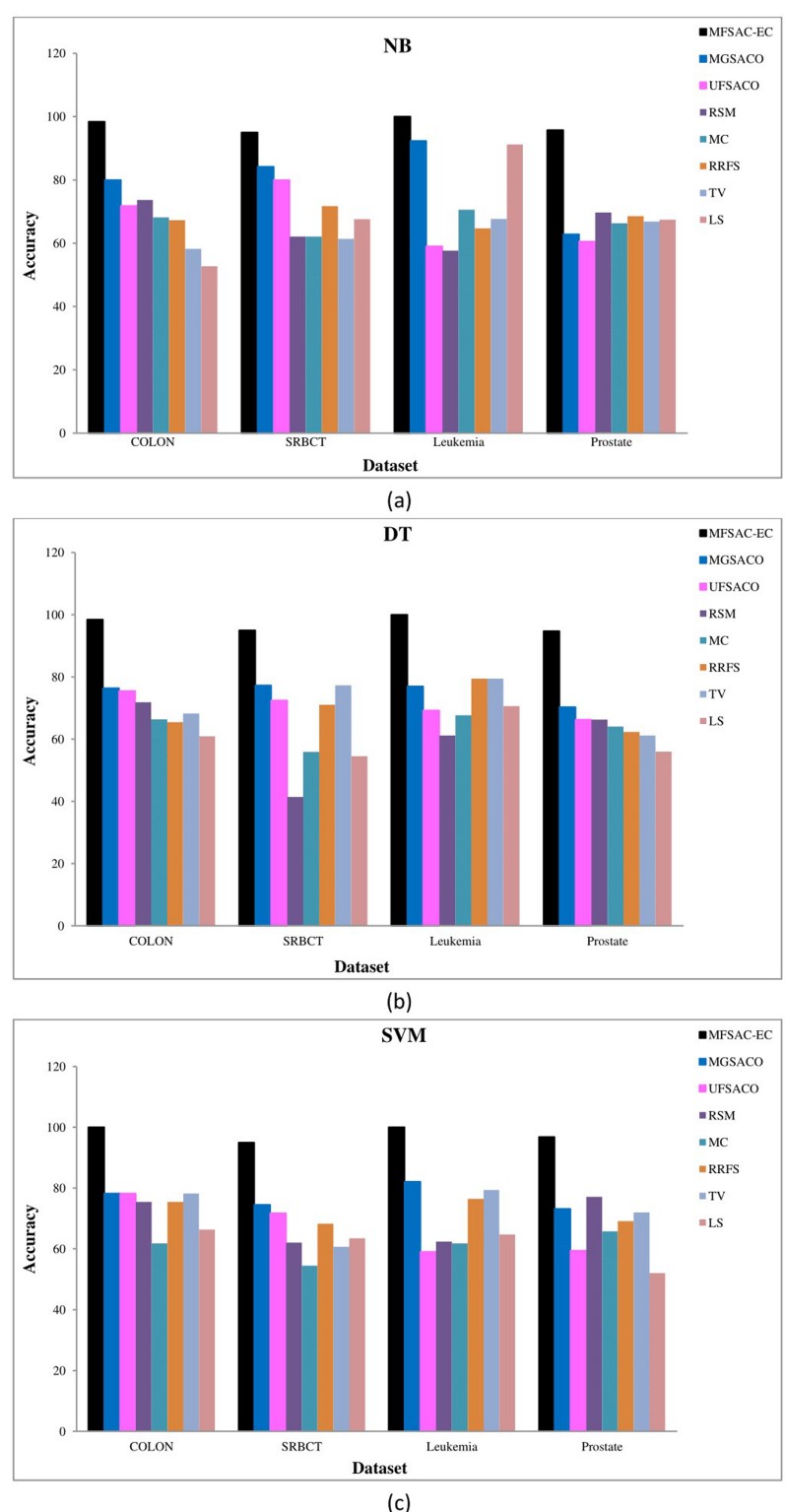

**Figure 7 Comparison of MFSAC-EC with other well-known unsupervised gene selection methods in terms of random splitting for different datasets.** Classification accuracy (%) of the MFSAC-EC model along with other unsupervised gene selection methods for four datasets are represented with different colored bars using (A) NB (B) DT and (C) SVM as base classifier.

**Table 9 Comparison of MFSAC-EC + DT with different existing Ensemble classifiers using DT in terms of tenfold cross validation.** Here MFSAC-EC + DT model is compared with existing ensemble classifiers where DT is used as base classifier. C4.5 algorithm is used as DT.

| | MFSAC-EC | PCA-based RotBoost | ICA-based RotBoost | AdaBoost | Bagging | Arcing | Rotation Forest | EN-NEW1 | EN-NEW2 |
|---|---|---|---|---|---|---|---|---|---|
| Colon | 98.39 | 95.48 | 96.1 | 94.97 | 94.92 | 69.35 | 95.21 | 79.03 | 83.87 |
| Leukemia | 100 | 98.75 | 98.77 | 98.22 | 97.47 | Not Found | 97.97 | Not Found | Not Found |
| Breast | 100 | 94.39 | 97.88 | 98.89 | 92.74 | 80.41 | 98.6 | 94.85 | 95.88 |
| Lung | 100 | 98.11 | 99.54 | 96.3 | 97.08 | 97.24 | 97.56 | 98.34 | 99.45 |
| Prostate | 97.79 | Not Found | Not Found | 90.44 | 94.12 | 87.5 | Not Found | 94.85 | 97.06 |
| MLL | 100 | 98.86 | 99.31 | 97.63 | 97.11 | 91.67 | 97.61 | 93.06 | 98.61 |
| SRBCT | 100 | 99.5 | 99.59 | 98.16 | 96.46 | Not Found | 97.44 | Not Found | Not Found |

**Table 10 Comparison of MFSAC-EC using DT, KNN, NB, SVM with different existing Ensemble classifiers using DT, KNN, NB, SVM in terms of tenfold cross validation.** Here classification accuracy (%) of four ensemble classifiers MFSAC-EC + NB, MFSAC-EC + KNN, MFSAC-EC+DT, and MFSAC-EC+SVM are shown with respect to results of other existing ensemble classifiers with the same base learners. The best accuracy (%) for every dataset is shown in bold.

| Dataset | MFSAC-EC | | | | Bagging | | | Boosting | | | Stacking | | | HBSA | | SD_Ens | Meta_Ens |
|---|---|---|---|---|---|---|---|---|---|---|---|---|---|---|---|---|---|
| | DT | NB | KNN | SVM | DT | NB | KNN | DT | NB | KNN | DT | NB | KNN | KNN | SVM | | |
| Leukemia | **100** | **100** | **100** | **100** | 94.12 | 88.23 | 73.53 | 91.18 | 88.24 | 75.53 | 91.18 | 91.18 | 91.18 | 88.46 | 88.46 | 92.45 | 94.12 |
| Colon | 98.39 | 98.39 | **100** | 98.39 | 95.16 | 66.13 | 90.32 | 98.39 | 87.1 | 91.94 | 98.39 | 93.59 | 93.59 | 75 | 85 | 94.4 | 99.21 |
| Prostate | 97.79 | **99.26** | **99.26** | **99.26** | 26.47 | 26.47 | 38.24 | 26.47 | 26.47 | 52.94 | 26.47 | 26.47 | 52.94 | 85.29 | 97.06 | 52.94 | 52.94 |
| Lung | **100** | **100** | **100** | **100** | 91.28 | 96.64 | 97.32 | 81.88 | 95.3 | 97.99 | 97.99 | 97.99 | 96.64 | Not Found | Not Found | 81.88 | 97.99 |
| Breast | **100** | **100** | **100** | **100** | 78.95 | 36.84 | 68.42 | 68.42 | 36.84 | 68.42 | 68.42 | 68.42 | 68.42 | Not Found | Not Found | 73.49 | 79.87 |

**Table 11 Comparison of MFSAC-EC using SVM and KNN with respect to different existing deep learning classifiers using random splitting.** Here classification accuracy (%) of two ensemble classifiers MFSAC-EC + KNN, and MFSAC-EC+SVM are shown with respect to results of other existing ensemble classifiers with the same base learners. The best accuracy (%) for every dataset is shown in bold.

| Dataset | SVM | | | KNN | | |
|---|---|---|---|---|---|---|
| | MFSAC-EC | Folded Autoencoder | Autoencoder | MFSAC-EC | Folded Autoencoder | Autoencoder |
| Colon | **100** | 90.15 | 73.11 | **98.39** | 81.09 | 56.97 |
| Prostate | **96.81** | 84.16 | 64.3 | **97.87** | 76.48 | 52.1 |
| Leukemia | **100** | 93.62 | 84.12 | **100** | 85.24 | 77.13 |

proposed MFSAC-EC classification model is superior to the existing ensemble classification models in almost every case. The superior performance of the proposed model is due to the following reasons:

- The generation of the different bootstrapped versions of training data and also the use of the oversampling procedure to balance the cardinality of majority class and minority class in every bootstrapped dataset reduces the chances of the overfitting problem of a classifier.

- Different types of filter methods are used in the MFSAC method. It has been already observed that one filter gives better performance for one dataset while the same gives poor results for other datasets. This is because every filter uses separate metrics and so the choice for a filter for a specific dataset is a very complex task. As different filter methods are used in the MFSAC method, so different sub-datasets with different characteristics-based attributes/genes are formed from each dataset. This is shown using Venn diagram in Figs. S4A and S4B. Here for Leukemia and Prostate cancer datasets, the first twenty genes, selected by each filter are shown. In case of Leukemia dataset, Relief measure generates non-overlapping gene subset while using other filter metrics presence of a small number of overlapping genes in different gene subsets are observed. In Prostate cancer dataset, Relief generates non-overlapping gene subset and also maximum number of genes are non-overlapping in gene subsets formed by Fisher score, MI (mutual information). From these figures, it is clear that using different filter methods different subsets of genes are selected and different sub-datasets are formed. It shows diversity of those filter methods. As a consequence, the base classifiers prepared on these diverse datasets are become diverse. This diversity increases the power of ensemble classifier.

- Moreover, the genes selected by different filter methos are good biomarker also. In Table 12, the top ranked eight genes selected by MFSAC-EC model are shown for Leukemia and Colon cancer datasets. Among these genes, gene MPO (with column number 1,720), CST3 (with column number 1,823), ZYX (with column number 4,788), CTSD (with column number 2,062), CD79A/MB-1(with column number 2,583), LYZ (with column number 6,738) in Leukemia dataset are important biomarkers as these are selected by different filter methods mentioned in Fig. S4.

- In MFSAC, at first, a sub-dataset of the most relevant genes is selected by each filter method. Then on each sub-dataset, the proposed supervised gene clustering algorithm is applied and a reduced sub-dataset of modified attributes/features in the form of augmented cluster representatives is generated. In this method, at the time of cluster formation, genes are augmented based on their supervised information. In other words, such augmentation is considered where it increases the class discrimination power. Thus effectively, the class relevance of any augmented cluster representative is greater than that of any single gene involved in that process. So, this modified sub-dataset containing a reduced feature set in the form of augmented cluster representatives is more powerful according to class discrimination power than the sub-dataset containing a subset of the most relevant genes. Apart from this, it is well known fact in gene expression data that two genes are functionally similar if they are pattern-based similar (either positively co-expressed or negatively co-expressed) (*Das et al., 2016*). So, at the time of the augmentation procedure, two types of augmentations are considered here. One is that a gene is added with its original value with the current cluster representative and another one is that the gene is added with its sign-flipped value with the current cluster representative. This is because if the current cluster representative and a gene are positively co-expressed then normal addition is considered but if they are

**Table 12 List of genes selected by MFSAC-EC model for the colon and leukemia cancer datasets.**

| Dataset | Gene name | Accession number | Description | Validation of genes |
|---|---|---|---|---|
| Colon | TPM1 | Hsa.1130 | Human tropomyosin isoform mRNA, complete cds. | *Gardina et al. (2006), Thorsen et al. (2008), Botchkina Inna et al. (2009)* |
| | IGFBP4 | Hsa.1532 | Human insulin-like growth factor binding protein-4 (IGFBP4) gene, promoter and complete cds. | *Durai et al. (2007), Singh et al. (1994), Yu & Rohan (2000)* |
| | MYL9 | Hsa.1832 | Myosin Regulatory Light Chain 2, Smooth Muscle Isoform (Human); contains element TAR1 repetitive element | *Yan et al. (2012), Zhu et al. (2019)* |
| | ALDH1L1 | Hsa.10224 | Aldehyde Dehydrogenase, Mitochodrial X Precursor (*Homo sapiens*) | *Feng et al. (2018), van der Waals, Borel Rinkes & Kranenburg (2018), Kozovska et al. (2018)* |
| | KLF9 | Hsa.41338 | Human mRNA for GC box binding protein/ Kruppel Like Factor 9, complete cds | *Brown et al. (2015), Ying et al. (2014), Simmen et al. (2008)* |
| | MEF2C | Hsa.5226 | Myocyte-Specific Enhancer Factor 2, Isoform MEF2 (Homosapiens) | *Chen et al. (2017), Giorgio, Hancock & Brancolinic (2018), Su et al. (2016)* |
| | GADPH | Hsa.1447 | Glyceraldehyde 3-Phosphate Dehydrogenase | *Zhang et al. (2015), Tang et al. (2012)* |
| | TIMP3 | Hsa.11582 | Metalloproteinase Inhibitor 3 Precursor | *Su et al. (2019), Bai et al. (2007)* |
| Leukemia | TXN | X77584_at | TXN Thioredoxin | *Kamal et al. (2016), Léveillard & Aït-Ali (2017), Karlenius & Tonissen (2010)* |
| | CSF3R | M59820_at | CSF3R Colony stimulating factor 3 receptor (granulocyte) | *Zhang et al. (2018), Ritter et al. (2020), Klimiankou et al. (2019), Lance et al. (2020)* |
| | MPO | M19508_xpt3_s_at | MPO from Human myeloperoxidase gene | *Szuber & Tefferi (2018), Kim et al. (2012), Lagunas-Rangel et al. (2017), Handschuh (2019)* |
| | LYZ | M21119_s_at | LYZ Lysozyme | *Wang et al. (2013), Liu et al. (2018), Tong & Ball (2014)* |
| | CST3 | M27891_at | CST3 Cystatin C (amyloid angiopathy and cerebral hemorrhage) | *Chen, Tsau & Lin (2010)* |
| | ZYX | X95735_at | Zyxin | *Chen, Tsau & Lin (2010), Qi & Yang (2013)* |
| | CTSD | M63138_at | CTSD Cathepsin D (lysosomal aspartyl protease) | *Wang et al. (2013)* |
| | CD79A/ MB-1 gene | U05259_rna1_at | MB-1 membrane glycoprotein | *Wang et al. (2013), Kozlov et al. (2005)* |

**Note:**
Here second column represents the gene names while third column indicate the gene accession number. The fourth column indicates the description of the gene while the fifth column indicates the literature where it has been referred as cancer biomarker.
The gene names and their corresponding accession numbers for both of the datasets COLON and LEUKEMIA can be found in the following links: COLON: http://genomics-pubs.princeton.edu/oncology/affydata/index.html, http://genomics-pubs.princeton.edu/oncology/affydata/names.html. LEUKEMIA: https://www.kaggle.com/crawford/gene-expression.

negatively co-expressed then normal addition will hamper the addition process and in that case, sign-flipping of that gene will give proper result. The effect of augmentation with respect to every filter method is shown in Fig. 8. In Fig. 8, for the Breast cancer dataset, at the time of supervised cluster formation from each filter generated subset, the original gene, and its corresponding class relevance value, and also augmented gene and its corresponding class relevance are shown. From Fig. 8, it is clear that for every filter method the class relevance score of every original gene is increased with respect to that filter after augmentation. In Fig. 8, different class labels are distinguished by different colors.

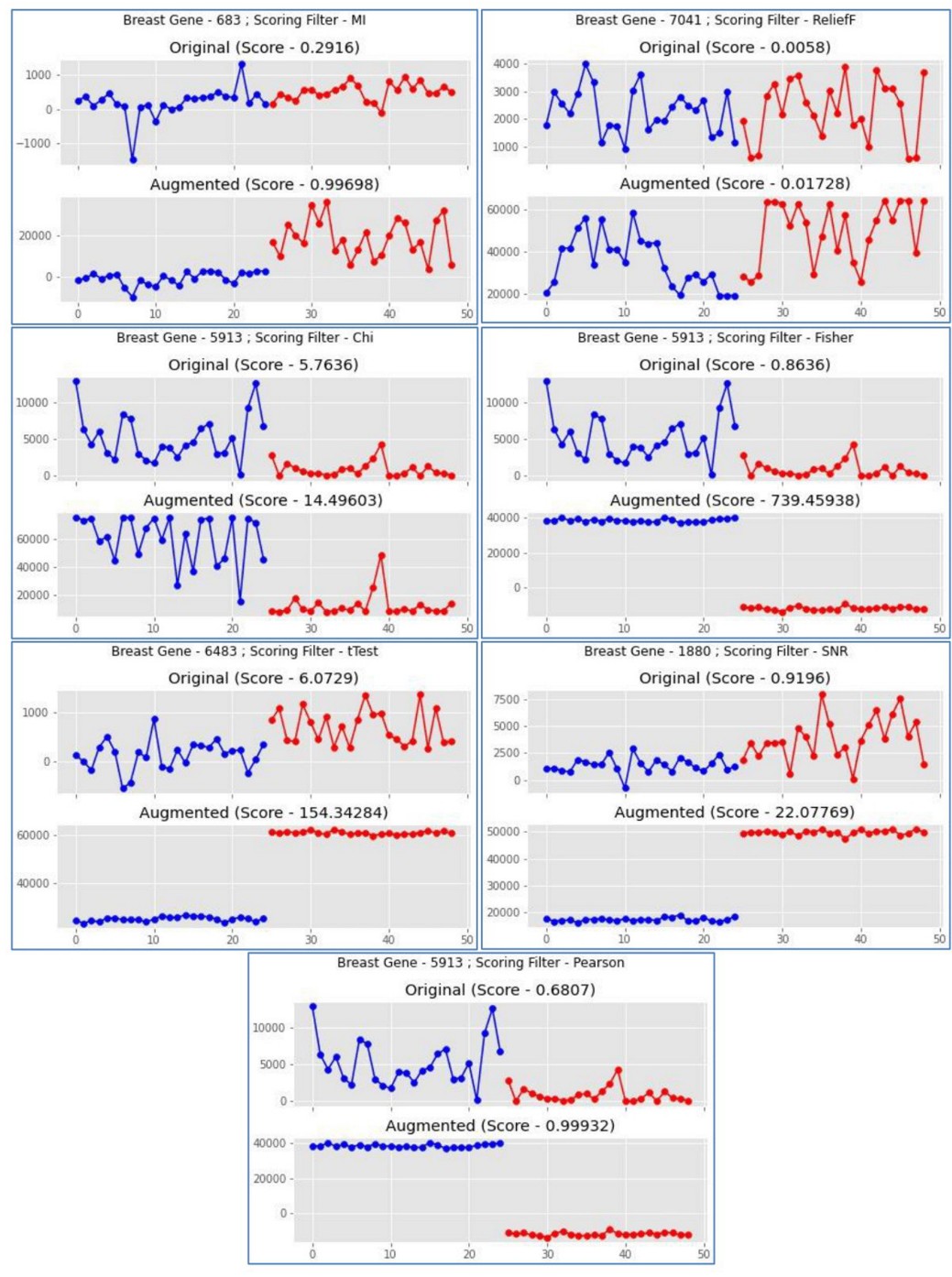

**Figure 8 Original gene (different class label with different color) and corresponding Augmented gene with respect to different filter methods for Breast Cancer dataset.** Seven figures for seven different filter score function are shown here. In each figure the original gene and augmented gene are plotted with respect to sample class label. X-axis represents class label while Y-axis represents expression value. Two different class labels are represented by blue and red color. The difference of expression values of two classes in the augmented gene shows class discrimination ability of that gene. Gene number is the column number in the original dataset.

- Finally, for each sub-dataset with modified attributes in the form of augmented cluster representatives, a classifier is constructed using any existing classifier, and these classifiers are combined using the majority voting technique to form an ensemble classifier (EC). The use of different sub-datasets with optimal gene subsets in the form of augmented cluster representatives and the formation of a classifier for every sub-dataset can solve the overfitting problem of any single classifier. This is due to the reason that not all sub-datasets can consistently perform well on all types of cancer datasets (due to inherent characteristics of the datasets), but due to the use of majority voting in ensemble classifiers, this problem can be solved or reduced.

Another outcome of our proposed model is to rank informative genes for every cancer dataset. For this task, the frequency of occurrence of each gene present in the form of augmented cluster representatives in every sub-dataset is counted and these genes are ranked according to the counted value to measure the importance of those genes for any specific disease, here cancer. To establish the biological significance of those selected genes for every cancer dataset, their contribution has been confirmed by other existing studies where they are referred already. From these existing studies, it is clear that the selected genes are important for cancer class discrimination and also are important as cancer biomarkers for molecular treatment targets.

## CONCLUSIONS

Many machine learning and statistical learning-based classifiers for sample classification already exist in the literature, but these methods are prone to suffer from overfitting due to small sample size problems, class imbalance problems, and the curse of the high dimensionality of microarray data. Although some of the existing methods can mitigate these issues to quite an extent, the problems have still not been satisfactorily overcome. Due to this reason, here a novel feature selection-based ensemble classification model named MFSAC-EC is proposed. It has been shown that the proposed model can handle the above-mentioned issues present in existing models. To check the performance of the proposed MFSAC-EC model, this classifier is applied to test sample classification accuracy in high dimensional microarray gene expression data, a domain that will be beneficial in the field of cancer research. From the experimental results, it has been found that the proposed model outperforms all other well-known existing classification models combined with the different recognized feature selection methods and also the newly developed ensemble classifiers for all types of cancer datasets mentioned here. Apart from this classification task, the proposed model can also rank informative attributes according to their importance. The efficiency of the proposed model in this task is vindicated by finding the most informative genes for the colon cancer and leukemia cancer datasets using this model. These genes are biologically validated based on other well-known existing studies. Consequently, it is clear that the selected genes are vital for sample class discrimination and are also important biomarkers for molecular treatment targets of deadly diseases.

### Funding

The authors received no funding for this work.

### Competing Interests

The authors declare that they have no competing interests.

### Author Contributions

- Shilpi Bose conceived and designed the experiments, performed the experiments, analyzed the data, performed the computation work, prepared figures and/or tables, authored or reviewed drafts of the paper, and approved the final draft.
- Chandra Das conceived and designed the experiments, performed the experiments, analyzed the data, performed the computation work, prepared figures and/or tables, authored or reviewed drafts of the paper, and approved the final draft.
- Abhik Banerjee performed the experiments, performed the computation work, prepared figures and/or tables, and approved the final draft.
- Kuntal Ghosh performed the experiments, analyzed the data, authored or reviewed drafts of the paper, and approved the final draft.
- Matangini Chattopadhyay analyzed the data, authored or reviewed drafts of the paper, and approved the final draft.
- Samiran Chattopadhyay analyzed the data, authored or reviewed drafts of the paper, and approved the final draft.
- Aishwarya Barik performed the experiments, prepared figures and/or tables, and approved the final draft.

### Data Availability

The code and datasets are available at GitHub:

https://github.com/NSECResearchCD-SLB/PEERJ_MFSAC_EC.

### Supplemental Information

Supplemental information for this article can be found online at http://dx.doi.org/10.7717/peerj-cs.671#supplemental-information.

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
