# Peer review of "An ensemble machine learning model based on multiple filtering and supervised attribute clustering algorithm for classifying cancer samples"

_PeerJ Computer Science, doi:10.7717/peerj-cs.671_

## Round 0.1 · original submission · Major Revisions

After reviewing the paper and the reviews, this paper requires Major Revision. The authors need to pay close attention to the comments of the reviewers; the manuscript needs editing/rewriting for sense, flow, coherence, cohesion, and readability. The authors should add a clearer rationale for the motivation of the paper in the introduction section and review the literature more thoroughly. In the introduction the authors should describe the contribution of the paper in detail and clearly state how and why it solves a gap in the field and advances knowledge. The procedures need to be written more clearly, so that the methods section can be easily replicable, readable, and understood. The reviewers provide valuable insights to the authors that will improve the manuscript as well as the accuracy and reproducibility of the results.

Reviewer 1 ·

Basic reporting

The draft is very preliminary. It is very verbose as some similar sentences/concepts were repeated several times, and a better organization is required to make it simple and clear.

The code is available through github.

Experimental design

The overall design is OK, but more details were required.

Validity of the findings

For the validation purpose, the authors should provide an independent dataset of the same cancer, using the genes selected by their filters on their current dataset, and then evaluate the classification performance on the independent dataset. In this case, their models were more general.

Additional comments

Review of Manuscript 54940, "An Ensemble Machine Learning Model based on Multiple Filtering and Supervised Attribute Clustering Algorithm for Classifying Cancer Samples "

This paper applied machine learning methods to classify cancer samples based on the gene expression profiles. Their method was an ensemble classification model based on 7 sets of genes from 7 different filters, namely: Fisher-score, Chi-square, Relief, t-test, Mutual information, SNR and Pearson correlation. Briefly, for a microarray gene expression dataset of U samples, and V genes, a filterScore was calculated for each gene for each filter. Then, for each filter, a greedy algorithm was used to choose the best subsets of genes that could best classify the samples. After that, data of a subset of genes were selected for one of 4 classical ML methods: K-Nearest Neighbor, Naive Bayes, Support vector machine, and Decision tree. Finally, a vote from all 7 filters concludes the class of a sample.

The draft is very preliminary. It is very verbose as some similar sentences/concepts were repeated several times, and a better organization is required to make it simple and clear.

Main comments:
1.Please add a summary table of the 7 filters in this paper. That table should include the definitions of each filter, and how the FT scores were calculated. In the current setting, the reviewer was confused about the differences between t-test and Chi-square, as most time, these 2 methods contain the same information when there are 2 classes.
2.The figure legends were missing.
3. Please use “gene”, or “variate”, or “feature” consistently throughout the draft.
4. From line 309 to 323, for each variable, please note whether they were matrixes, vectors, or just genes.
5. Why not just tell the readers Y=7, to reduce symbols in the draft.
6. Line 341-343:
“Compute two augmented representatives 〖AR〗_xk^+ and 〖AR〗_xk^- by adding G_j and its complement with 〖AR〗_xk respectively
Compute class relevance value of 〖AR〗_xk^+ and 〖AR〗_xk^- using 〖FT〗_x score function”
Please write clearly how those values were calculated.
7. Line 250: 〖GS〗_x=〖GS〗_x⁄G_m , Is this updating GSx values? From Line 315,”〖GS〗_x is a subset of top-ranked genes of G selected using 〖FT〗_x score function and SD_x is corresponding sub dataset.”, GSx is a set of genes. Really confusing.
8. A venn diagram of genes selected for each filters could help understand the distinct contributions of each filter for each dataset.

Reviewer 2 ·

Basic reporting

There are some nonstandard quotations of references. Such as lines 59,61,67,75,96, 99, 102 and so on. The references 5,10,63 and so on.

Experimental design

The authors proposed a supervised algorithm for classifying cancer samples. It will be reasonable if they compare it with some supervised algorithms instead of unsupervised algorithms.

Validity of the findings

no comment

Additional comments

This manuscript proposed a machine learning based on a new ensemble classification model named MFSAC-EC for classifying cancer samples and gene selection. They used some classifiers and data sets for comparison, and obtained better results. Though novel, such minor improvement may not lead to novel biomarker discovery. There are still some questions. The language needs significant corrections as there are many grammatical errors and typos.
Below are specific questions, suggestions, and comments.
(1) The authors proposed a supervised algorithm for classifying cancer samples. It will be reasonable if they compare it with some supervised algorithms instead of unsupervised algorithms.
(2) How is the data imbalance addressed? Using accuracy as an evaluation metric may be misleading given that the datasets are commonly imbalanced in cancer research.
(3) There are some nonstandard quotations of references. Such as lines 59,61,67,75,96, 99, 102 and so on. The references 5,10,63 and so on.
(4) More information about feature selection should be detailed in the Introduction.
(5) There is some redundant information. In the Results, such as lines 430-432; 434-437; 451-454 and so on. Please check the whole manuscript.
(6) Figure 4 is very hard to read, please consider enlarging the figure and improve the resolution.

Reviewer 3 ·

Basic reporting

- A very complex ensemble learning based method is proposed for cancer classification on microarray data and identification of important genes for classification
- Achieved very good classification accuracy (90~100%) on 10 public datasets with limited number of samples (maximum is 181 samples)

Experimental design

- The proposed MFSAC-EC method consists of two stages [See Figure 3]
Stage 1: create sub-datasets and select associated genes to be used for classifier creation in stage 2

> Step 1: use 7 filter score functions ((1) Fisher-score (2) Chi-square (3) Relief (4) t-test (5) Mutual information (6) SNR and (7) Pearson Correlation) to select sub-datasets and associated genes
> Step 2: use supervised attribute/gene clustering algorithm (SAC) to refine select sub-datasets and associated genes
>Step 3: identify most important genes for classification by counting occurrence frequency of genes in step 2

Stage 2: ensemble multiple classifiers using sub-datasets from stage 1
> (training) Use sub-datasets and associated genes form step 2 of stage 1 to train 4 types of classifiers ((1) K-Nearest Neighbor, (2) Naive Bayes, (3) Support vector machine, (4) and Decision tree)
> (apply) Use majority voting to ensemble result from 4 classifiers

Validity of the findings

- Achieved very good classification accuracy (90~100%) on 10 public datasets with limited number of samples (maximum is 181 samples)
> Performed better than other well-known ensemble learning classifiers [table 7, 8, Figure 8]
> Performed better than other well-known gene selection methods [Figure 7]

Additional comments

- Method is very complex (does it need to be so complex in order to perform well?) and there are too many options and parameters to adjust for performance optimization. For example, there are many choices of score functions and ensembled classifiers. And, how to adjust associated parameters of those filter score functions and classifiers.
> No processing time comparison data. Ensemble method could take much longer to implement and execute.
> Superior performance is claimed on small data samples.

- Authors should consider unsupervised methods and potentially deep learning as it has been shown numerous times that it can be much more efficient at generalizing over large data pools. At least the authors should discuss why they did not use unsupervised methods or deep learning.

- [Line 265-272] Description of merging process and its connection with Figure 2 is not easy to understand.
> “Here in the case of gene1, the augmented1+ gives the better score while for gene2 augmented2- gives a better relevance score.”
- The better score for gene1 and better relevance score of gene2 are not provided.

- Figure 3. No differentiation between data block and processing block in the block diagram. They are both shown as rectangular shape blocks. To help the reader, they could be shown in different shape of blocks.
> Figure 2, Figure 4 (invisible), Figure 9
- Charts without axis titles or invisible axis titles

---

## Round 0.2 · accepted · Accept

After reviewing the re-submitted manuscript and the one review, I concur with the one reviewer to accept the resubmitted manuscript. The re-submitted manuscript needs still some minor editing for sense, flow, coherence, and cohesion. Overall, it is an important study and should be considered for publication in PeerJ. Comp. Sci. once these minor issues have been resolved.

Reviewer 1 ·

Basic reporting

In the revised manuscript, the authors addressed all my concerns.

Experimental design

no comment

Validity of the findings

no comment

Additional comments

In the revised manuscript, the authors addressed all my concerns.